# Chronic Ca²⁺ imaging of cortical neurons with long-term expression of GCaMP-X

Jinli Geng[1,2†], Yingjun Tang[3†], Zhen Yu[1,2†], Yunming Gao[1,2], Wenxiang Li[1,2], Yitong Lu[1,2], Bo Wang[1,2], Huiming Zhou[1,2,3], Ping Li[1], Nan Liu[4], Ping Wang[5], Yubo Fan[1], Yaxiong Yang[1*], Zengcai V Guo[3*], Xiaodong Liu[1,2*]

[1]Advanced Innovation Center for Biomedical Engineering, School of Biological Science and Medical Engineering, School of Engineering Medicine, Key Laboratory for Biomechanics and Mechanobiology of Ministry of Education, Beihang University, Beijing, China; [2]X-Laboratory for Ion-Channel Engineering, Beihang University, Beijing, China; [3]Tsinghua-Peking Joint Center for Life Sciences, IDG/McGovern Institute for Brain Research, School of Medicine, Tsinghua University, Beijing, China; [4]Center for Life Sciences, School of Life Sciences, Yunnan University, Kunming, China; [5]Key Laboratory for Biomedical Engineering of Ministry of Education, Zhejiang University, Hangzhou, China

*For correspondence:
yangyaxiong@buaa.edu.cn (YY);
guozengcai@tsinghua.edu.cn
(ZVG);
liu-lab@buaa.edu.cn (XL)

†These authors contributed
equally to this work

Competing interest: The authors
declare that no competing
interests exist.

Reviewing Editor: Henry M
Colecraft, Columbia University,
United States

**Abstract** Dynamic Ca²⁺ signals reflect acute changes in membrane excitability, and also mediate signaling cascades in chronic processes. In both cases, chronic Ca²⁺ imaging is often desired, but challenged by the cytotoxicity intrinsic to calmodulin (CaM)-based GCaMP, a series of genetically-encoded Ca²⁺ indicators that have been widely applied. Here, we demonstrate the performance of GCaMP-X in chronic Ca²⁺ imaging of cortical neurons, where GCaMP-X by design is to eliminate the unwanted interactions between the conventional GCaMP and endogenous (apo)CaM-binding proteins. By expressing in adult mice at high levels over an extended time frame, GCaMP-X showed less damage and improved performance in two-photon imaging of sensory (whisker-deflection) responses or spontaneous Ca²⁺ fluctuations, in comparison with GCaMP. Chronic Ca²⁺ imaging of one month or longer was conducted for cultured cortical neurons expressing GCaMP-X, unveiling that spontaneous/local Ca²⁺ transients progressively developed into autonomous/global Ca²⁺ oscillations. Along with the morphological indices of neurite length and soma size, the major metrics of oscillatory Ca²⁺, including rate, amplitude and synchrony were also examined. Dysregulations of both neuritogenesis and Ca²⁺ oscillations became discernible around 2–3 weeks after virus injection or drug induction to express GCaMP in newborn or mature neurons, which were exacerbated by stronger or prolonged expression of GCaMP. In contrast, neurons expressing GCaMP-X were significantly less damaged or perturbed, altogether highlighting the unique importance of oscillatory Ca²⁺ to neural development and neuronal health. In summary, GCaMP-X provides a viable solution for Ca²⁺ imaging applications involving long-time and/or high-level expression of Ca²⁺ probes.

## Editor's evaluation

This paper addresses the toxicity of fluorescent calcium indicators, comparing two series of indicators (GCaMPs and GCaMP-Xs) in mouse cortical neurons. Focusing on calcium oscillations in relation to neuronal morphology, the paper documents GCaMP side effects following prolonged and/or strong expression, and establishes that GCaMP-X indicators are less toxic both in vitro and in vivo. The paper will be of interest to neuroscientists (and others) who use fluorescence calcium indicators for chronic Ca²⁺ imaging.

## Introduction

$Ca^{2+}$ signals play pivotal roles in the brain, closely involved in membrane excitability, sensory transduction, synaptic transmission, neural development, and plasticity (*Berridge et al., 2003*). $Ca^{2+}$ dysregulations are linked with the mental disorders including Parkinson's diseases, Alzheimer's diseases, epilepsy, and schizophrenia (*Chan et al., 2007*; *Fernández de Sevilla et al., 2006*; *Khan et al., 2020*; *Liebscher et al., 2016*), suggesting that multiple factors and cascades may converge to $Ca^{2+}$ as one of the central factors underlying brain diseases, referred as the calcium hypothesis (*Berridge, 2010*). According to their downstream consequences, cellular $Ca^{2+}$ signals could be categorized as genomic versus non-genomic, to reflect the fact that in some cases where gene expressions are regulated (chronic) versus in other cases only the (acute) functions of existing proteins are concerned.

Acutely, cellular $Ca^{2+}$ reflects single-neuron activities, such as spontaneous fluctuations and stimulus-evoked responses (*Chen et al., 2013*; *O'Banion and Yasuda, 2020*). $Ca^{2+}$ imaging is often utilized to measure neuronal excitability, as one alternative to electrical recording. In fact, genetically encoded $Ca^{2+}$ indicators (GECIs) represented by GCaMP, based on CaM (calmodulin) and $Ca^{2+}$/ CaM-binding motif M13, have been broadly applied to monitor neurons and other excitable cells (*Akerboom et al., 2012*; *Chen et al., 2013*; *Dana et al., 2019*; *Nakai et al., 2001*; *Tallini et al., 2006*; *Tian et al., 2009*; *Yang et al., 2018*). In addition to a faithful index of acute responses (in the timescale of seconds/minutes, such as a burst of action potentials), $Ca^{2+}$ is often tightly coupled to various chronic effects or processes, for example, $Ca^{2+}$-dependent gene transcription and expression, neurite outgrowth or pruning, long-term potentiation or depression, learning and memory, and neural degeneration (*O'Banion and Yasuda, 2020*). Therefore, it is highly desirable to monitor the long-term $Ca^{2+}$ dynamics (days/weeks or longer) for cells, tissues, organs, or even whole organisms, which would greatly facilitate mechanistic understanding of the genomic/chronic roles of $Ca^{2+}$ in diverse pathophysiology (*Garcia et al., 2017*). Meanwhile, back to the context of non-genomic $Ca^{2+}$, longitudinal imaging may have a broad scope of applications where long-term changes in responses or behaviors are of interest. In parallel with $Ca^{2+}$ imaging, different types of electrodes, such as MEA (multiple electrode array) and flexible electronics, have been extensively deployed for cultured neurons, brain slices, live animals, or human brains to record neural activities in the format of neuronal action potentials, local-field potentials, and EEG (*Hong and Lieber, 2019*). The goal is to monitor neural activities across multiple days, weeks, or even the entire lifespan in the studies of training/behaviors, retina/ vision, brain disorders, addictions, and pharmacological and interventional therapeutics (*Aramuni and Griesbeck, 2013*; *Couto et al., 2021*). GECIs hold great promise to avoid chronic immune responses and recoding instability that electrodes are often encountered with (*Aramuni and Griesbeck, 2013*). Indeed, GCaMP and other GECIs have been demonstrated as more advantageous methods over electrodes or dyes during chronic imaging of neurons (*Aoki et al., 2017*; *Murphy et al., 2020*; *Tian et al., 2009*). Unfortunately, neural toxicities often accompany long-term expression of GCaMP or chronic GCaMP imaging with either virus infected (*Chen et al., 2013*; *Tian et al., 2009*; *Yang et al., 2018*) or transgenic neurons (*Steinmetz et al., 2017*).

Perturbing L-type $Ca_V1$ channels and presumably other (apo)CaM-binding proteins, GCaMP indicators cause side-effects in neurons which have been documented especially for enhanced or prolonged expressions. The unwanted molecular events of GCaMP may or may not be manifested as the toxic effects at the cellular or system levels. For instance, when employing viral infection, imaging experiments are restricted within the empirical time window and dosage range to alleviate nuclear-filling of GCaMP (*Resendez et al., 2016*); or for transgenic mice, special promoters, conditional expression, and other tactics are utilized to reduce the expression level/time of GCaMP (*Madisen et al., 2015*). With such work-around solutions, rich information and rapid progress in neurosciences have been achieved by GCaMP imaging. At a cost, special cautions and procedures are required due to GCaMP toxicity, for example virus dilution trials (*Resendez et al., 2016*), which often cause inconvenience in experiments besides other potential problems (e.g., nucleus-filling or low SNR).

Under the testing conditions in this work, neuronal perturbations were evidenced from GCaMP: with either earlier or newer versions, for either viral or transgenic expression, either in vitro (cultured neurons) or in vivo (living mice), and on either acute sensory response to whisker deflection or spontaneous oscillation encoding genomic $Ca^{2+}$. GCaMP-X has been designed to resolve these perturbations by eliminating unwanted interferences with endogenous (apo)calmodulin signaling. For in vivo imaging beyond the safe time window (3-week extension) and dosage (10-fold higher), GCaMP-X

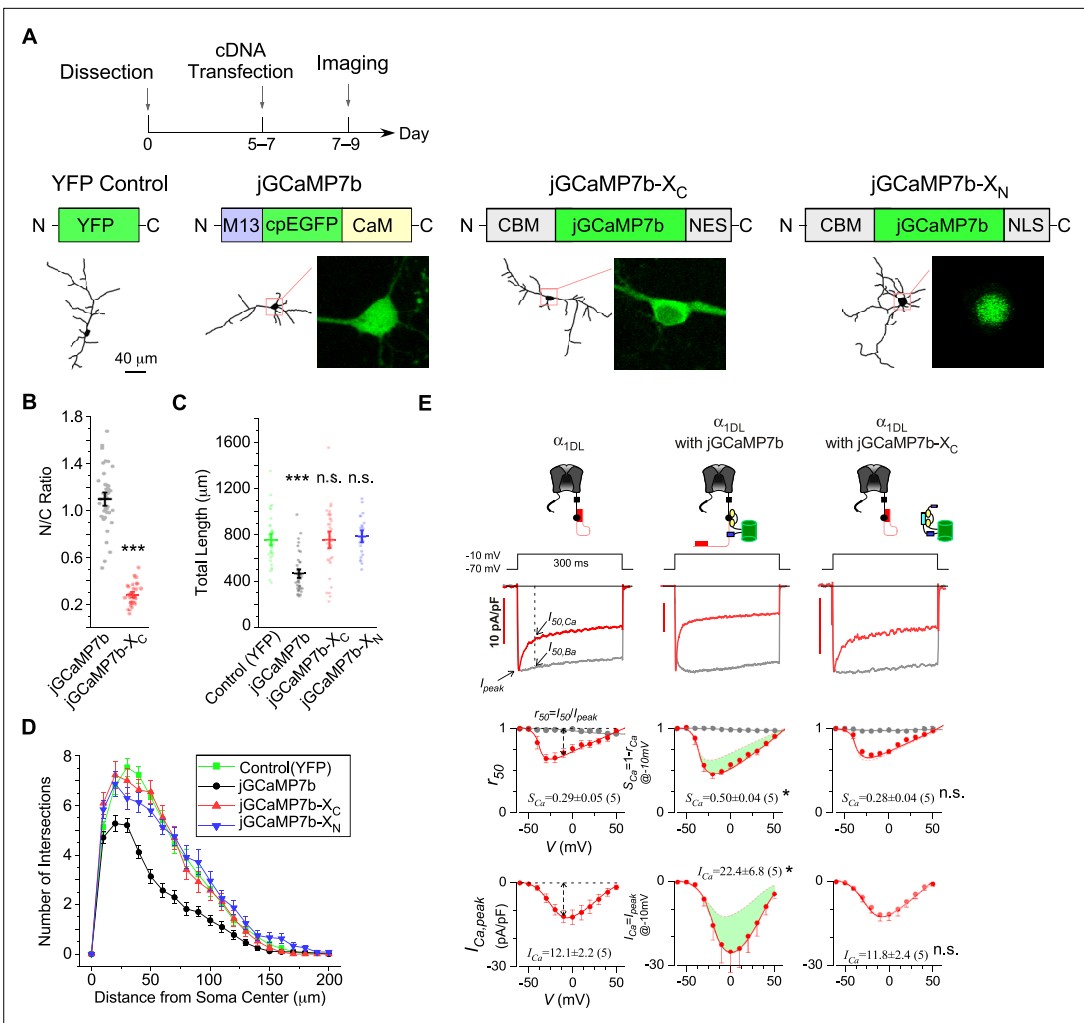

**Figure 1.** The design principles applicable to jGCaMP7 and jGCaMP7-X. (**A**) Cultured cortical neurons from newborn mice were transiently transfected with YFP, jGCaMP7b, jGCaMP7b-X$_C$, or jGCaMP7b-X$_N$, respectively, on DIV 5–7, then imaged by confocal microscopy on DIV 7–9. As illustrated, apo<u>C</u>aM-<u>b</u>inding <u>m</u>otif (CBM) was fused onto N-terminus of GCaMP and the tags of localization signals (nuclear <u>e</u>xport <u>s</u>ignal or nuclear <u>l</u>ocalization <u>s</u>ignal, NES/NLS) were fused to the C-terminus of GCaMP to construct GCaMP-X$_C$ or GCaMP-X$_N$, respectively. Neurite tracing and subcellular GCaMP distributions are shown below. (**B**) N/C ratio of jGCaMP7b or jGCaMP7b-X$_C$ in neurons, by calculating the nucleus versus cytosol ratio of fluorescence intensities. Total length (**C**) and *Sholl* analysis (**D**) for cortical neurons expressing YFP or GCaMP variants. Two independent experiments from two independent culture preparations (**A–D**). (**E**) Electrophysiological validations of jGCaMP7b versus jGCaMP7b-X$_C$ with recombinant Ca$_V$1.3 channels. Full-length Ca$_V$1.3 channels ($\alpha_{1DL}$) were expressed in HEK293 cells alone (left) or with jCaMP7b (middle) or with jGCaMP7b-X$_C$ (right). At −10 mV, Ca$^{2+}$ current traces (red, with scale bars indicating current amplitudes) and Ba$^{2+}$ current traces (gray, rescaled) are shown. $S_{Ca}$ and $I_{Ca}$ (quantified by the equations shown in the first column) are the indices of calcium-dependent inactivation and voltage-dependent activation, respectively. Cell numbers are indicated in the parentheses right after the values. Standard error of the mean (SEM) and two-tailed unpaired Student's *t*-test (**B**) or one-way analysis of variance (ANOVA) followed by Bonferroni for post hoc tests (**C, E**) (criteria of significance: \*p < 0.05; \*\*\*p < 0.001; *n.s.* denotes 'not significant') were applied.

The online version of this article includes the following source data and figure supplement(s) for figure 1:

**Figure supplement 1.** Expression levels of indicators in HEK293 cells.

**Figure supplement 1—source data 1.** Source data for *Figure 1—figure supplement 1A*.

**Figure supplement 1—source data 2.** Source data for *Figure 1—figure supplement 1C*.

**Figure supplement 2.** Expression levels of indicators in cultured cortical neurons.

**Figure supplement 3.** Differential interactions of GCaMP versus GCaMP-X with apoCaM-binding proteins.

**Figure supplement 3—source data 1.** Source data for *Figure 1—figure supplement 3A, B*.

outperformed GCaMP in recapitulating both sensory responses to whisker stimulation and autonomous $Ca^{2+}$ fluctuations. In cultured cortical neurons, GCaMP of strong and/or prolonged expression caused the damage to neurites accompanied by aberrant $Ca^{2+}$ oscillations, all overcome by GCaMP-X as a simple solution, which also highlights the importance of oscillatory $Ca^{2+}$ to neurons both in vitro and in vivo.

## Results

### Design principles of GCaMP-X validated by newer GCaMP versions

Recently, GCaMP has been updated to its newest versions jGCaMP7 (but also see the bioRxiv preprint of jGCaMP8 [*Zhang et al., 2021*]), with enhanced sensing performance in multiple aspects over the previous GCaMP6 (*Dana et al., 2019*; *Grødem et al., 2021*). Considering that the design of jGCaMP7 is also on the basis of CaM, we postulated that jGCaMP7-contained apoCaM would have similar problems to those of earlier GCaMPs. We chose jGCaMP7b to further validate the design principles of neuron-compatible GCaMP-X established in our previous work (*Yang et al., 2018*). Following the protocol of transient transfection (*Figure 1A*), jGCaMP7b accumulated into the nuclei in a substantial subpopulation of cortical neurons indexed by N/C (nucleus/cytosol) ratio (*Figure 1B*), considered as the hallmark of GCaMP side-effects. Notably, jGCaMP7b exhibited even more severe nuclear accumulation than other GCaMP variants, which may account for the nuclear jGCaMP7b evidenced in vivo (*Dana et al., 2019*). Accordingly, the total length (*Figure 1C*) and the complexity (*Figure 1D*) of neurites were significantly reduced in jGCaMP7b-expressing neurons. The apoCaM-binding motif (CBM) and the localization tags were then appended onto jGCaMP7b, following the design of GCaMP-X (*Yang et al., 2018*), to construct jGCaMP7b-$X_C$ and jGCaMP7b-$X_N$ for cytosolic and nuclear $Ca^{2+}$ imaging, respectively. GCaMP-X is supposed to eliminate its binding to apoCaM targets in neurons and reduce the cytotoxicity intrinsic to GCaMP, a critical issue to long-term $Ca^{2+}$ monitoring. Depicted by neurite tracing (*Figure 1A*), both cytosolic and nuclear versions of jGCaMP7-X have greatly enhanced the compatibility with neurons. In fact, neurons expressing jGCaMP7-X were essentially indistinguishable from GFP control neurons, in direct contrast to the neurons transfected with jGCaMP7 of the same amount of cDNA (*Figure 1C, D*). In light of electrophysiological analyses on calmodulation of $Ca_V1$ channels (*Ben-Johny and Yue, 2014*; *Yang et al., 2018*), we examined the effects of jGCaMP7b on recombinant $Ca_V1.3$ channels in HEK293 cells. jGCaMP7b significantly altered the major properties of $Ca_V1.3$ gating, that is, both inactivation and activation were enhanced (*Figure 1E*). The expression level is a critical factor to evaluate the side-effects of GCaMP. We then examined the actual levels of expressed proteins in HEK293 cells when transiently transfected with the same amount of cDNA for different indicators (*Figure 1—figure supplement 1*). Demonstrated by both western blotting and immunohistochemistry, jGCaMP7b in HEK293 cells was expressed either in the cytosol or in the nucleus at the same levels as jGCaMP7b-$X_C$ or jGCaMP7b-$X_N$, respectively. Likewise, jGCaMP7 and jGCaMP7-X were at the same cytosolic or nuclear levels of expression when transiently transfected into neurons (*Figure 1—figure supplement 2*). Moreover, by coexpressing jGCaMP7b-$X_C$ and jGCaMP7b-$X_N$, the total expression level estimated by immunostaining was even higher than that of jGCaMP7b, the latter of which caused significant damage to neurites whereas the total neurite length of jGCaMP7b-X neurons was about the same as GFP control neurons. Therefore, the expression levels of GCaMP-X versus GCaMP did not underlie their differential performances.

Notably, by coimmunoprecipitation GCaMP bound $Ca_V1.3$ ($\alpha_{1D}$) at its apoCaM-binding domain whereas GCaMP-X was found no binding (*Figure 1—figure supplement 3*). Similarly, neurogranin, an important postsynaptic apoCaM-binding protein (*Gerendasy and Sutcliffe, 1997*), was unveiled to bind GCaMP but not GCaMP-X. Thus, the above direct evidence of molecular interactions further consolidated the design principles of GCaMP-X, serving as the major candidate mechanism of cellular GCaMP toxicities.

### Acute sensory responses monitored by viral expression of GCaMP-X in vivo

GECIs including GCaMP have been widely applied to monitor neuronal responses to various stimuli. Due to the cytotoxicity known from the very early versions of GCaMP, in vivo imaging experiments are normally required to conduct within the time window. In practice, an optimal time window (OTW)

is about 3–8 weeks postinjection for GCaMP-infected neurons of live mice (*Chen et al., 2013*; *Huber et al., 2012*; *Resendez et al., 2016*), in order to achieve substantial levels of GCaMP expression and fluorescence but not too high levels prone to side-effects. Here, we investigated into the $Ca^{2+}$ dynamics under whisker stimulation within or beyond OTW by applying the adeno-associated viruses (AAVs) of GCaMP6m or GCaMP6m-$X_C$ with the neurospecific *Syn* promoter to S1 primary somatosensory cortex in the brain of adult mice (2-month age or older) (*Figure 2A*). To exclude the potential bias due to level of expression, GCaMP6m-$X_C$ viruses ($1.0 \times 10^{13}$ v.g./ml) of higher dose than GCaMP6m ($1.0 \times 10^{12}$ v.g./ml) were injected (60 nl/injection). Progressive nuclear accumulation of GCaMP was previously reported in vivo and in vitro (*Chen et al., 2013*; *Yang et al., 2018*; *Zariwala et al., 2012*). Consistently, by the criteria of N/C ratio (0.8), nucleus-filling GCaMP was observed in a fraction of neurons 4–6 weeks postinjection, and the average N/C ratio was substantially increased when examined 8–13 weeks postinjection (*Figure 2B*). In direct contrast, no neuron expressing GCaMP6m-$X_C$ fell into the nucleus-filled category even weeks beyond OTW (up to 13 weeks postinjection). In the earlier study (*Chen et al., 2013*), the impairment on visual responses was reported from nucleus-filled neurons after long-term expression of GCaMP6 (several months postinjection); and during the initial phase (1- to 2-month postinjection) nuclear GCaMP6 did not perturb the proper physiology of neurons. Here, nucleus-filled neurons expressing GCaMP6m (N/C ratio >0.8) started to show less responsiveness than neurons expressing GCaMP6m-$X_C$ as early as within OTW (4–6 weeks) (*Figure 2A*), which may underlie the lower amplitude of $Ca^{2+}$ responses ($\Delta F/F_0$) averaged over the whole population of neurons (*Figure 2C*). Another two indices of success rate and SNR (signal to noise ratio) were also consistent with the above notion that within OTW neurons filled with nuclear GCaMP might have been impaired. Beyond OTW (8–13 weeks), GCaMP6m was more frequently and clearly found in the nucleus, and the neurons exhibited more significant differences from GCaMP6m-$X_C$ according to all the three indices of $\Delta F/F_0$, success rate, and SNR.

Our data have extended the advantages of GCaMP-X over GCaMP from in vitro onto in vivo. Collectively, neurons may suffer from GCaMP side-effects either within or beyond OTW, for which nuclear GCaMP expression appears to be a critical factor. GCaMP-X outperformed GCaMP in imaging sensory-evoked $Ca^{2+}$ dynamics especially beyond OTW, which provides a simple but promising solution for long-term $Ca^{2+}$ imaging.

## Long-term monitoring of $Ca^{2+}$ oscillations by GCaMP-$X_C$ in vitro

Before proceeding further with in vivo GCaMP-X imaging, we decided to conduct in-depth examinations on the long-term performance of GCaMP-X under in vitro conditions. Of note, slow $Ca^{2+}$ oscillations have been observed from a variety of excitable or non-excitable cells (*Uhlén and Fritz, 2010*). In neurons, oscillatory $Ca^{2+}$ signals can increase the efficiency and specificity of gene expression (*Dolmetsch et al., 1998*; *Li et al., 1998*), thus playing important role in neuronal functions, development, morphology, and even general health (*Kamijo et al., 2018*; *Nicotera and Orrenius, 1998*; *Toth et al., 2016*). Spontaneous electrical activities are initiated at the early stage of neural development, and subsequently become more synchronized (*Luhmann et al., 2016*; *Spitzer, 2006*). Such longitudinal processes of oscillatory $Ca^{2+}$ and neuritogenesis which are mechanistically coupled may serves as a perfect scenario to demonstrate the side-effects of GCaMP on neurons. We then hypothesized that GCaMP-X versus GCaMP neurons would make a clear difference in their chronic recordings of oscillatory $Ca^{2+}$ signals. Meanwhile, GCaMP-X with enhanced neuron compatibility is expected to provide new insights into the roles of spontaneous $Ca^{2+}$ oscillations in neuronal morphology (*Gomez and Zheng, 2006*; *Rosenberg and Spitzer, 2011*). To monitor such $Ca^{2+}$ dynamics in vitro, the AAVs carrying GCaMP6m or GCaMP6m-$X_C$ (for in vitro use) were equally (1 μl, $1 \times 10^{12}$ v.g./ml) added to the cortical neurons of neonatal mice (DIV 0, 0-day in vitro) which were maintained and examined till DIV 28 (*Figure 3A, B*). Fluctuations of $Ca^{2+}$ activities were perceivable starting from the first week (DIV 3 and DIV 6) with GCaMP6m-$X_C$, in the pattern of high-frequency, low-amplitude, and unsynchronized signals. On DIV 10, the oscillation frequency decreased while the amplitude was increased with an enhanced level of synchronization. On DIV 28, $Ca^{2+}$ oscillations of individual or subgrouped neurons became widely synchronized featuring robust spikes and slow frequency (*Figure 3—video 1*), indicative of the formation of neural circuitry. In contrast, $Ca^{2+}$ signals were severely distorted in GCaMP6m-infected DIV-10 neurons. Despite that the performance of GCaMP6m in the first week resembled GCaMP6m-$X_C$, longer expression time of GCaMP6m resulted in altered patterns of $Ca^{2+}$ oscillations

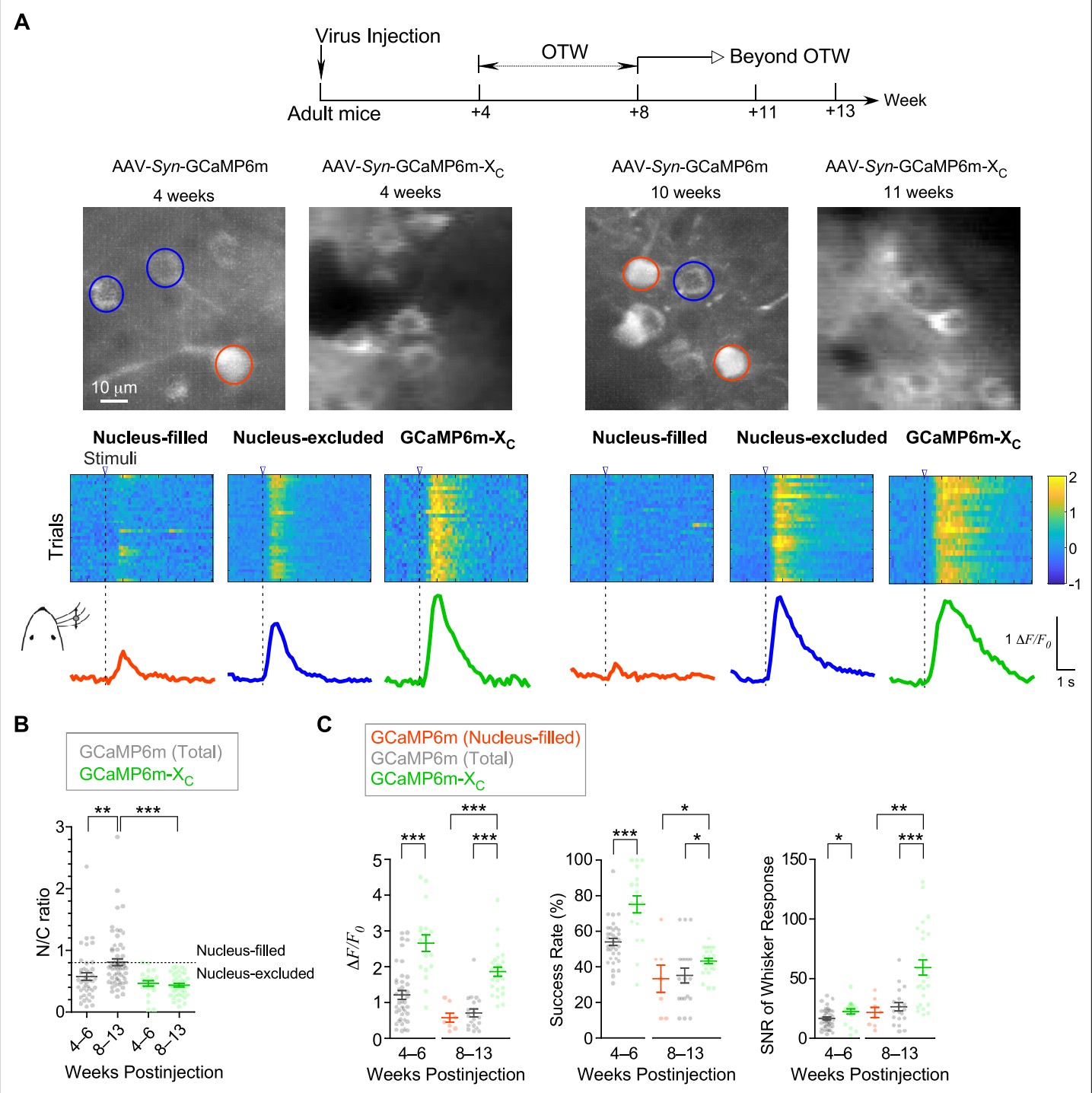

**Figure 2.** In vivo $Ca^{2+}$ imaging of sensory-evoked responses in cortical neurons virally infected with GCaMP-X versus GCaMP. (**A**) $Ca^{2+}$ dynamics of GCaMP6m and GCaMP6m-$X_C$ in S1 primary somatosensory cortex under whisker stimulation by in vivo two-photon $Ca^{2+}$ imaging. 4–6 and 8–13 weeks postinjection were considered as optimal time window (OTW) or beyond OTW, respectively. The orange and blue circles in the representative two-photon images indicate nucleus-filled and nucleus-excluded GCaMP6m, respectively. The colored scale bar indicates the fluorescence intensity of $Ca^{2+}$ probes. Multiple trials of the same neuron are shown for nucleus-filled GCaMP6m, nucleus-excluded GCaMP6m, or GCaMP6m-$X_C$. The averaged $Ca^{2+}$ responses are shown at the bottom. (**B**) Subcellular distributions of GCaMP6m and GCaMP6m-$X_C$ in vivo within OTW or beyond OTW. Neurons were divided into two distinct subgroups, nucleus-excluded and nucleus-filled, by applying the cutoff value (N/C ratio) of 0.8. (**C**) Responses to repetitive whisker stimuli were evaluated and compared for GCaMP6m nucleus-filled neurons, total GCaMP6m neurons, and GCaMP6m-$X_C$. The average amplitude ($\Delta F/F_0$) (left), success rate of the trials (middle) and SNR (right) during whisker stimuli were compared. Data were obtained from five

*Figure 2 continued on next page*

*Figure 2 continued*

or four mice for GCaMP6m and GCaMP6m-$X_C$, respectively. Standard error of the mean (SEM) and one-way analysis of variance (ANOVA) followed by Bonferroni for post hoc tests (criteria of significance: *p < 0.05; **p < 0.01; ***p < 0.001) were calculated when applicable.

(*Figure 3—videos 2 and 3*). One major abnormality was the substantial reduction in oscillatory activities of GCaMP6m-expressing neurons, which was manifested after DIV 10 by much longer intervals between two adjacent peaks and much smaller amplitudes in average. Occasionally, abnormal $Ca^{2+}$ spikes with ultralong lasting duration could be observed on DIV 17 (*Figure 3B*, *Figure 3—video 2*). Around DIV 28, cease of $Ca^{2+}$ oscillations and broken neurites were often evidenced (*Figure 3—video 3*). We then further analyzed oscillatory $Ca^{2+}$ signals by the frequency and other key indices across the timespan from DIV 3 up to DIV 28. Statistical results demonstrated that the frequency of $Ca^{2+}$ fluctuation with GCaMP6m-$X_C$ was about 150 mHz during the first week, then gradually declined to the plateau around 20 mHz (*Figure 3C*). Meanwhile, the peak amplitude exhibited a rising trend in the neurons expressing GCaMP6m-$X_C$ across the full term (*Figure 3D*). In contrast, both the frequency and the amplitude of $Ca^{2+}$ oscillations acquired by GCaMP6m were drastically changed after DIV 17, and then even more deteriorated later in that the oscillation was less and less recognizable and eventually halted on DIV 28 (*Figure 3C, D*). Synchronization is one major hallmark of autonomous $Ca^{2+}$ oscillations, which was evaluated by the mean of correlation coefficient. As demonstrated by the temporal profiles of correlation coefficients, the comparison between GCaMP6m versus GCaMP6m-$X_C$ unveiled a crucial phase turning from increasing to decreasing synchronization around DIV 17–21 in GCaMP6m-expressing neurons (*Figure 3E*). Likewise, the full width at half maximum (FWHM), another index of oscillatory waveforms, was aberrantly wider for GCaMP6m than GCaMP6m-$X_C$, becoming noticeable on DIV 10, and much more pronounced (10-fold) later on (*Figure 3F*). Collectively from these indices, GCaMP indeed caused progressive damage on cortical neurons along with the culturing time or developmental stages; in contrast, GCaMP6m-$X_C$ has overcome nearly all the above negative effects, emerging as a promising tool for chronic $Ca^{2+}$ imaging with enhanced neural compatibility. Also, Fast Fourier Transformation (FFT) was applied to the $Ca^{2+}$ waveforms acquired by GCaMP6m-$X_C$ (*Figure 3G* and *Figure 3—figure supplement 1*). The distribution of frequency components started to change during DIV 10–17, when slow $Ca^{2+}$ oscillations of 10–100 mHz appeared to be the dominant form (*Figure 3H*). Based on separate preparations of neurons, we conducted another two experiments to compare GCaMP6m and GCaMP6m-$X_C$ up to DIV 42 (*Figure 3—figure supplement 2*). Statistical results from these data on frequency, $\Delta F/F_0$, correlation coefficient per view and FWHM support that GCaMP-X outperforms GCaMP in chronic $Ca^{2+}$ imaging of cultured neurons (*Figure 3I*). The newer probes of jGCaMP7b and jGCaMP7b-$X_C$ resulted in differential performance that jGCaMP7b-$X_C$ was much less toxic, consistent with the above notion (*Figure 3—figure supplement 3*).

In summary, the same set of neurons were chronically monitored with GCaMP-X across the development stages from newborn to mature, by which the temporal profiles of the major characteristics were achieved for oscillatory $Ca^{2+}$ signals in cultured cortical neurons. As one additional control, Fluo-4 AM ($Ca^{2+}$ dye) imaging was conducted for synchronized $Ca^{2+}$ oscillations on DIV 21 (*Figure 3—figure supplement 4*). The key parameters of $Ca^{2+}$ dynamics measured by jGCaMP7b-$X_C$ were much closer to those by Fluo-4 AM than jGCaMP7b, supporting that jGCaMP7b-$X_C$ is less toxic than jGCaMP7b. Minor to moderate differences still existed between Fluo-4 AM and jGCaMP7b-$X_C$, which were more likely attributed to intrinsic probe properties (e.g., $Ca^{2+}$-sensing kinetics) rather than neuronal toxicities.

## Close correlations between autonomous $Ca^{2+}$ oscillations and neuronal morphology in vitro

Spontaneous $Ca^{2+}$ oscillations, the slow periodic $Ca^{2+}$ waveforms in particular, are tightly coupled with neuronal morphology, development and neuritogenesis (*Kamijo et al., 2018*; *Toth et al., 2016*). GCaMP-X promises unprecedented opportunities for concurrent imaging of both neuronal functionalities and morphogenesis across different stages of development. Such chronic $Ca^{2+}$ imaging is difficult to implement if using other approaches, for example, conventional GCaMP or $Ca^{2+}$ dyes, both would cause side-effects to neurons (*Smith et al., 2018*; *Yang et al., 2018*). By taking advantage of GCaMP-X, we here aimed at the relationship between cellular $Ca^{2+}$ and neuronal morphology.

Indistinguishable from control neurons infected with GFP viruses (*Figure 4—figure supplement 1*), neurons expressing GCaMP6m-$X_C$ followed the typical development process of neonatal neurons,

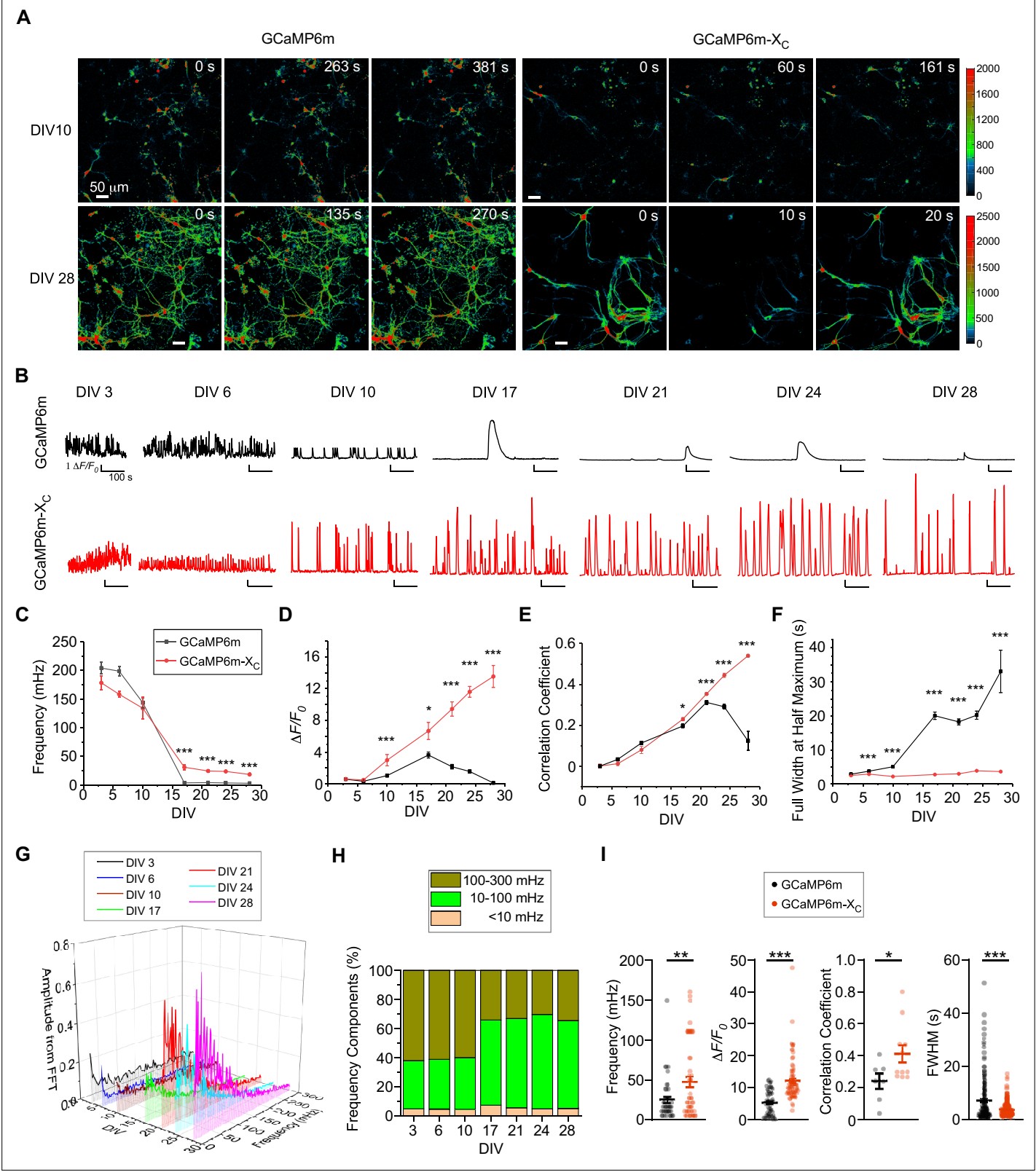

**Figure 3.** Chronic Ca²⁺ fluorescence imaging for autonomous Ca²⁺ oscillations in cultured cortical neurons. (**A**) Time-lapse images of cultured cortical neurons infected with AAV-*Syn*-GCaMP6m or AAV-*Syn*-GCaMP6m-X_C. Spontaneous Ca²⁺ activities by the two probes are shown for DIV 10 and DIV 28. See *Figure 3—videos 1–3* for details. Ca²⁺ signals (color coded) were monitored by the confocal microscope with a live-cell imaging chamber to maintain the cell culture conditions (37°C, 5% CO₂, 97–100% humidity) at different timepoints. (**B**) Representative traces of Ca²⁺ activities in cultured

*Figure 3 continued on next page*

*Figure 3 continued*

cortical neurons expressing GCaMP6m (upper) or GCaMP6m-X$_C$ (lower) from DIV 3 to DIV 28. Temporal profiles of key indices measured from spontaneous Ca$^{2+}$, including the average frequency ($10^{-3}$ Hz or mHz) (**C**) and peak amplitude ($\Delta F/F_0$, **D**); synchrony (quantified by the mean of correlation coefficient per view, **E**) and full width at half maximum (FWHM, **F**). (**G**) Power spectral analyses by FFT (Fast Fourier Transformation) for Ca$^{2+}$ traces of cortical neurons from DIV 3 to DIV 28. (**H**) Frequency components in percentage. By integrating the absolute amplitudes over each frequency band (**G**), three major bands are shown: <10 mHz (ultra-slow), 10–100 mHz (slow), and 100–300 mHz (fast), where the 10–100 mHz band indicates the major component. (**I**) Summary over three independent experiments with three independent culture preparations (see *Figure 3—figure supplement 2* for the other two experiments). Key indices of frequency (DIV 35), $\Delta F/F_0$ (DIV 28), correlation coefficient per view (DIV 28) and FWHM (DIV 28) were calculated and compared for GCaMP6m versus GCaMP6m-X$_C$. Standard error of the mean (SEM) and the Student's *t*-test (two-tailed unpaired with criteria of significance: *p < 0.05; **p < 0.01; ***p < 0.001) were calculated when applicable.

The online version of this article includes the following video and figure supplement(s) for figure 3:

**Figure supplement 1.** Spectral analysis for Ca$^{2+}$ waveforms acquired by GCaMP-X.

**Figure supplement 2.** Ca$^{2+}$ oscillations in long-term cultured cortical neurons expressing GCaMP6m or GCaMP6m-X$_C$ in vitro.

**Figure supplement 3.** Ca$^{2+}$ oscillations in cultured cortical neurons expressing jGCaMP7b or jGCaMP7b-X$_C$.

**Figure supplement 4.** Ca$^{2+}$ oscillations in cultured cortical neurons imaged by Fluo-4 AM.

**Figure 3—video 1.** Spontaneous Ca$^{2+}$ oscillation of cultured cortical neurons virally expressing GCaMP6m-X$_C$ on DIV 28.

https://elifesciences.org/articles/76691/figures#fig3video1

**Figure 3—video 2.** Spontaneous Ca$^{2+}$ oscillation of cultured cortical neurons virally expressing GCaMP6m on DIV 17.

https://elifesciences.org/articles/76691/figures#fig3video2

**Figure 3—video 3.** Spontaneous Ca$^{2+}$ oscillation of cultured cortical neurons virally expressing GCaMP6m on DIV 28.

https://elifesciences.org/articles/76691/figures#fig3video3

including neurite elongation/arborization and soma enlargement (*Figure 4A*). In contrast, these developmental processes were severely impaired by virally delivered GCaMP6m, especially after DIV 14 onwards, when nuclear accumulation and neurite shortening became evident. Depicted by DIV-28 neurons with neurite tracings, GCaMP6m caused significant damage on neurite outgrowth, and to the extreme, discernable death of neurons, in contrast to GCaMP6m-X$_C$ which had no apparent perturbation. In addition, the temporal profiles across the full time-course were achieved for both GCaMP6m (*Figure 4B*) and GCaMP6m-X$_C$ (*Figure 4C*) by the major indices of neurite length and soma size. At the early phase (before DIV 17), no significant difference between the two groups of GCaMP6m versus GCaMP6m-X$_C$ could be detected. However, toward the late stage (DIV 28 or later) of GCaMP-expressing neurons, the soma size was as small as ~200 μm$^2$ in contrast to the neurons expressing GFP or GCaMP6m-X$_C$ (~300 μm$^2$), as confirmed by the statistical summary over three independent experiments (*Figure 4—figure supplement 1*). Likewise, the total neurite length of GCaMP-expressing neurons rapidly declined, whereas GCaMP-X-expressing neurons went through an initial phase (~2 weeks) of rapid outgrowth before entering into the plateau phase, forming a monotonic increasing curve. Similar to neuritogenesis, the temporal profile of soma size also exhibited an upward-plateau trend (*Figure 4C*). Combining the data and analyses from both developmental and functional perspectives (*Figures 3 and 4*), we speculated on the potential correlations between neuronal growth and spontaneous Ca$^{2+}$ activities (*Figure 4—figure supplement 2*). Functionally, Ca$^{2+}$ dynamics appeared to be either ascending (amplitude) or descending (frequency) along with the developmental stages (DIV) (*Figure 4D*). Roughly, the oscillation amplitude linearly ($R^2 = 0.84$) correlated with the neurite length in total (*Figure 4—figure supplement 2A*). In direct contrast, the oscillation frequency and the total neurite length were inversely correlated ($R^2 = 0.99$) (*Figure 4—figure supplement 2B*). Resembling the amplitude, the level of synchrony (correlation coefficient) indicative of circuitry formation was positively correlated with the total neurite length ($R^2 = 0.86$) (*Figure 4—figure supplement 2C*). All these tight correlations support the notion that spontaneous Ca$^{2+}$ activities including its mature form of synchronized Ca$^{2+}$ oscillations may underpin neuritogenesis (*Estrada et al., 2006*; *Gomez and Zheng, 2006*; *Kamijo et al., 2018*). According to the spectral analyses (*Figure 3G, H*), the frequency band of 10–100 mHz played a crucial role in Ca$^{2+}$-dependent neuritogenesis.

Previous studies mainly relied on measuring transient Ca$^{2+}$ and neurite growth rate within a brief period of time (*Ito et al., 2010*; *Mukai et al., 2003*; *Rosenberg and Spitzer, 2011*; *van Pelt et al., 2004*). However, the overall neurite outgrowth across the developmental phases may help elucidate the roles of Ca$^{2+}$ in neuritogenesis, which has been lacking due to the difficulties of long-term Ca$^{2+}$

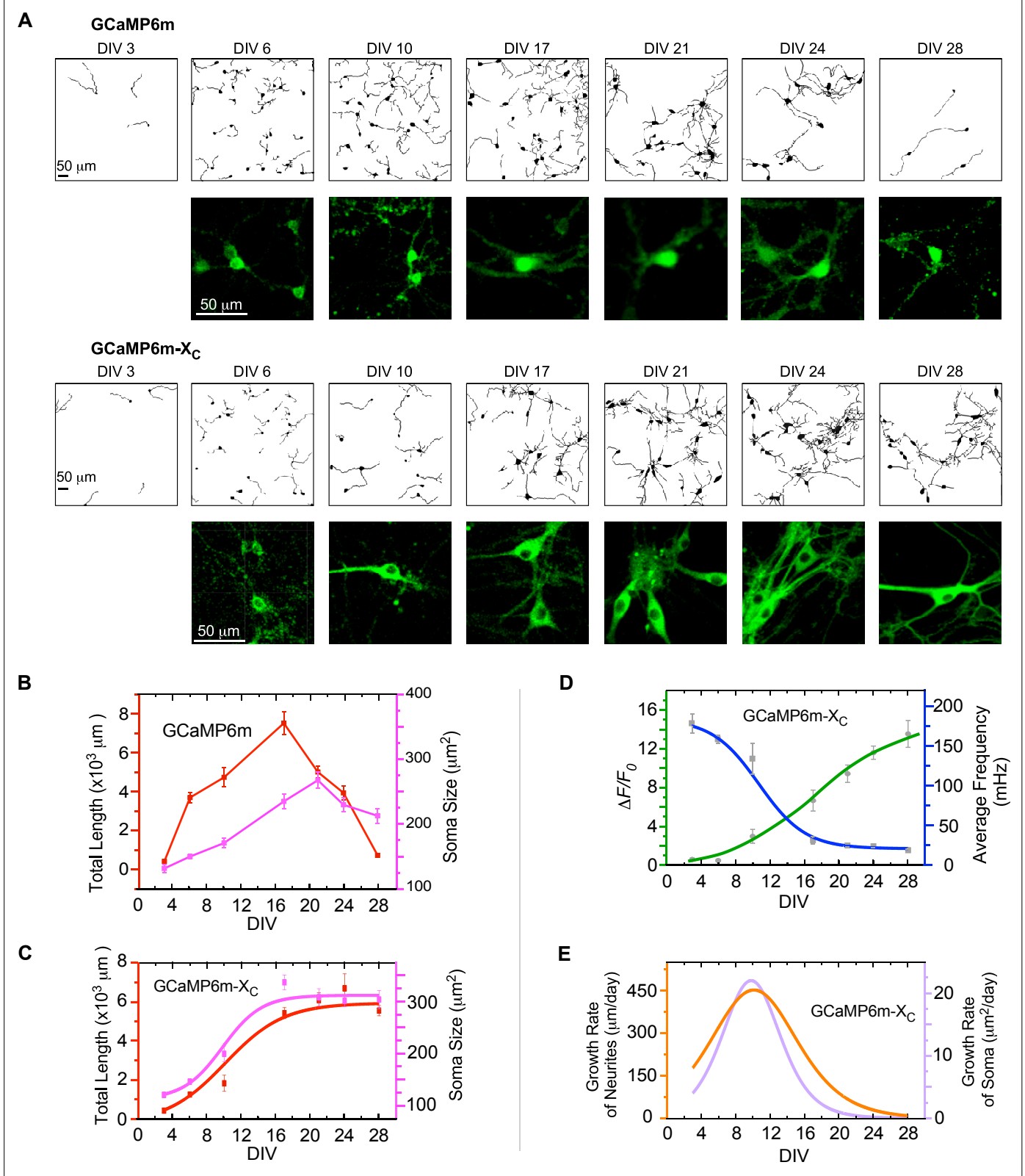

**Figure 4.** The correlations between neuronal development and Ca²⁺ oscillations unveiled by GCaMP-X. (**A**) Time-lapse images with neurite tracing for cultured cortical neurons expressing GCaMP6m (upper two rows) or GCaMP6m-X_C (lower two rows). Enlarged fluorescence images are to show subcellular distributions of the probes, indicative of the nuclear accumulation of GCaMP6m versus GCaMP6m-X_C. (**B**) Temporal profiles of total neurite length per view (red) or soma size per neuron (pink) for neurons expressing GCaMP6m. (**C**) Temporal profiles of total neurite length per view (red) or

*Figure 4 continued on next page*

*Figure 4 continued*

soma size per neuron (pink) for neurons expressing GCaMP6m-$X_C$. (**D**) The temporal profiles of the average frequency of $Ca^{2+}$ oscillations (blue) and the peak amplitude ($\Delta F/F_0$, green), adopted from *Figure 3C, D*. (**E**) Temporal profiles of the growth rates (µm/day) of neurite length (orange) or soma size (purple), respectively. Analyses were performed in parallel on both $Ca^{2+}$ waveforms and neuronal morphology based on the data obtained from the same three independent culture preparations as in *Figure 3* (see *Figure 4—figure supplement 1* for details on morphology data). Standard error of the mean (SEM) was calculated when applicable.

The online version of this article includes the following source data and figure supplement(s) for figure 4:

**Figure supplement 1.** Indistinguishable neuronal morphology between GCaMP6m-$X_C$ and GFP of long-term expression.

**Figure supplement 2.** Potential relationships between and neurite length and oscillation characteristics.

**Figure supplement 3.** Morphological analysis to compare jGCaMP7b-$X_C$ and jGCaMP7b in cultured cortical neurons.

**Figure supplement 4.** The long-term effects of GCaMP-X with enhanced expression levels in cultured neurons.

**Figure supplement 4—source data 1.** Source data for *Figure 4—figure supplement 4A*.

imaging. In fact, contradictory observations have been reported regarding how $Ca^{2+}$ actually regulates the rates of neuronal growth. Here, based on GCaMP-X imaging data, the first derivative of the growth curves was calculated as the growth rate of neurites or soma (*Figure 4E*). The bell-shaped curves suggested that the relationships between $Ca^{2+}$ oscillations and neuronal growth rates may depend on developmental stages, which reached the peak rates around DIV 10 for both neurite length and soma size. Therefore, the maximum growth rate appeared to be determined by both amplitude and frequency of $Ca^{2+}$ oscillations. In general, there might not be a simple relationship between oscillation characteristics and neuronal development. Within a short or brief timeframe, the growth rates in relation to various combinations of amplitude and frequency could be complicated (*Gomez and Zheng, 2006*), especially without considering the development stage of neurons. Similar results were obtained from jGCaMP7b-$X_C$ in comparison with jGCaMP7b (*Figure 4—figure supplement 3*). In summary, chronic GCaMP-X imaging provided a glimpse of the potential roles of slow $Ca^{2+}$ oscillations in neuritogenesis across multiple stages of neuronal development.

To further exclude any potential artifact related to probe expressions, a gradient of expression levels by jGCaMP7b-$X_C$ viruses were examined in cultured cortical neurons (*Figure 4—figure supplement 4*). 3 µl AAV-*Syn*-jGCaMP7b-$X_C$ ($1.0 \times 10^{12}$ v.g./ml) and 1 µl AAV-*Syn*-jGCaMP7b ($1.0 \times 10^{12}$ v.g./ml) led to the similar levels of whole-cell expression (the former would express much more in the cytosol). Under such conditions, the results from the two groups of neurons were consistent with those with equal amounts/volumes of viruses. jGCaMP7b-$X_C$ was much less toxic than jGCaMP7b, by comparing the indices of neuronal growth and $Ca^{2+}$ oscillations (*Figure 4—figure supplement 4F*), where neurite length and soma size of neurons expressing high-level jGCaMP7b-$X_C$ were nearly indistinguishable from GFP control neurons.

## Chronic imaging of spontaneous $Ca^{2+}$ activities in vivo

GCaMP6m perturbed autonomous $Ca^{2+}$ oscillations, presumably as one leading cause of neuronal toxicities. Such tight correlations between $Ca^{2+}$ dysregulations and aberrant morphology were clearly manifested during early development, which may extend onto mature neurons. The viruses were added to cultured cortical neurons at the mature stage (DIV 21), which were subsequently examined to compare the effects of jGCaMP7b versus jGCaMP7b-$X_C$ (*Figure 5—figure supplement 1*). Analyses of both neurites and oscillations demonstrated similar side-effects of jGCaMP7b in comparison with jGCaMP7b-$X_C$, starting to show up on DIV 28 and later on DIV 35 exhibiting significant differences in neurite length and oscillation characteristics. Similar to cultured neurons, spontaneous $Ca^{2+}$ activities in vivo are also correlated to gene transcription and expression at the cellular and circuity levels (*Laviv et al., 2020*; *Takahashi et al., 2016*). Therefore, based on our experiments and other published reports, a common theme of correlation exists between spontaneous $Ca^{2+}$ and neuronal morphology, for both premature and mature neurons, and both in vitro and in vivo (*Figure 5—figure supplement 2*). For adult mouse brain infected by AAV viruses of GCaMP6m or GCaMP6m-$X_C$ (the same procedures and dosages as in *Figure 2*), we characterized spontaneous $Ca^{2+}$ activities in S1 primary somatosensory cortex (*Figure 5A, B* and *Figure 5—videos 1 and 2*). Two checkpoints were set at 4 weeks postinjection (within OTW) and at 8 or 11 weeks postinjection (prolonged expression time beyond OTW), respectively. Similar to whisker deflection-response experiments in *Figure 2*,

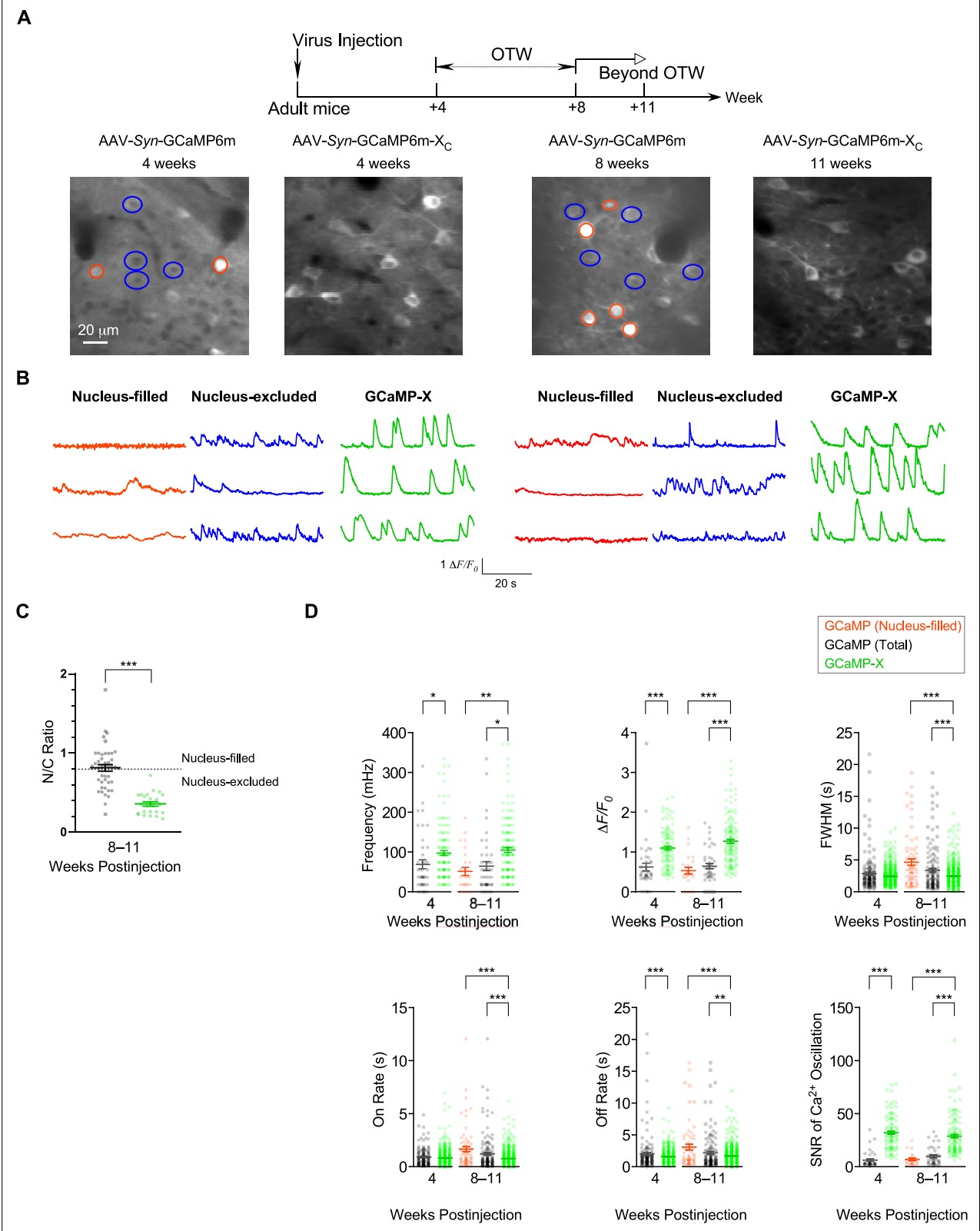

**Figure 5.** Chronic Ca²⁺ imaging for spontaneous Ca²⁺ activities in vivo. In vivo two-photon fluorescence images (**A**) and spontaneous nuclear Ca²⁺ activities (**B**) of virus-infected neurons in S1 primary somatosensory cortex. Neurons expressing GCaMP6m-X_C or GCaMP6m (nucleus-filled and nucleus-excluded) within OTW (optimal time window, 4–8 weeks postinjection) and beyond OTW (8–11 weeks postinjection) were analyzed and compared. (**C**) N/C ratio summary of GCaMP6m and GCaMP6m-X_C in vivo beyond OTW. Neurons were divided into two groups: nucleus-excluded and nucleus-filled,

*Figure 5 continued on next page*

*Figure 5 continued*

by applying the criteria of N/C ratio (0.8). (**D**) Key parameters of spontaneous $Ca^{2+}$ activities. Standard error of the mean (SEM) and two-tailed unpaired Student's *t*-test (**C**) or one-way analysis of variance (ANOVA) followed by Bonferroni for post hoc tests (**D**) (criteria of significance: *$p < 0.05$; **$p < 0.01$; ***$p < 0.001$) were calculated when applicable. Data were obtained from three or two mice for GCaMP6m and GCaMP6m-$X_C$, respectively.

The online version of this article includes the following video and figure supplement(s) for figure 5:

**Figure supplement 1.** Effects of virally expressed GCaMP versus GCaMP-X on mature cortical neurons.

**Figure supplement 2.** The critical timepoints of the protocols for both in vitro and in vivo experiments.

**Figure 5—video 1.** In vivo two-photon imaging of spontaneous $Ca^{2+}$ oscillation of neurons virally expressing GCaMP6m-$X_C$ in S1 primary somatosensory cortex beyond optimal time window (11 weeks postinjection).

https://elifesciences.org/articles/76691/figures#fig5video1

**Figure 5—video 2.** In vivo two-photon imaging of spontaneous $Ca^{2+}$ oscillation of neurons virally expressing GCaMP6m in S1 primary somatosensory cortex beyond optimal time window (8 weeks postinjection).

https://elifesciences.org/articles/76691/figures#fig5video2

the nucleus-filled neurons exhibited noticeable abnormalities in spontaneous $Ca^{2+}$ activities even within OTW. Beyond OTW, nucleus-filling was often found from GCaMP6m while very rare from GCaMP6m-$X_C$ as indexed by N/C ratio (*Figure 5C*). Accordingly, the damage was much exacerbated, as evidenced from the total or nucleus-filled neurons expressing GCaMP in comparison with the total neurons expressing GCaMP-X (*Figure 5D*). With GCaMP6m, the frequency and amplitude resulted in significantly lower values, accompanied by aberrantly wider FWHM and slower on/off rates. In contrast, neurons expressing GCaMP6m-$X_C$ maintained robust and stable spontaneous $Ca^{2+}$ activities with key characteristics within the normal ranges across the full term of experiments (up to 11 weeks postinjection). Notably, GCaMP6m-$X_C$ significantly improved the SNR calculated from spontaneous $Ca^{2+}$ signals in vivo, both within and beyond OTW (*Figure 5D*).

## Effects on neuronal morphology in vivo during long-term expression of GCaMP versus GCaMP-X

Similar to chronic GCaMP-X imaging in vitro, a similar correlation has been expected between oscillatory $Ca^{2+}$ and neuronal morphology for live adult mice which may underlie GCaMP side-effects observed from in vivo imaging (*Figures 2 and 5*). Firstly, titrations of GCaMP viruses were applied to characterize the dose-dependent damage of neurons, for which the concentrations were at $1 \times 10^{11}$, $5 \times 10^{11}$, and $1 \times 10^{12}$ v.g./ml for AAV-*Syn*-GCaMP6m, and $1 \times 10^{12}$ v.g./ml for AAV-*Syn*-GCaMP6m-$X_C$, respectively. The viruses (30 nl) at the above concentrations were microinjected into different brain regions of the same mouse and then after 3 weeks the expression levels in brain slices were examined (*Figure 6A*). Low-concentration injection of virus at $1 \times 10^{11}$ v.g./ml exhibited extremely sparse expression of GCaMP6m and yielded a low cell count. Correspondingly, the fluorescence signals were difficult to distinguish from the background, that is, low contrast and SNR. The virus concentration, when increased to $5 \times 10^{11}$ v.g./ml, resulted in a relatively larger number of healthy-looking cells expressing GCaMP6m. But the low image contrast still affected proper detection of $Ca^{2+}$ signals due to neuropil fluorescence. High-expression levels of GCaMP (at the virus concentration of $1 \times 10^{12}$ v.g./ml) significantly enhanced the fluorescence image contrast and greatly increased the numbers of GCaMP-positive cells. However, the majority of neurons exhibited severe nuclear accumulation, which would subsequently lead to aberrant $Ca^{2+}$ dynamics and cell death (*Figure 6B*). In contrast, high-dose injection of GCaMP6m-$X_C$ virus at $1 \times 10^{12}$ v.g./ml was beneficial for image contrast and the number of positive and healthy cells; meanwhile, the N/C ratio remained within the low range as expected. Next, under the conditions similar to *Figures 2 and 5*, we injected 60 nl GCaMP6m viruses of high dose ($1 \times 10^{12}$ v.g./ml) and GCaMP6m-$X_C$ viruses of ultrahigh dose ($1 \times 10^{13}$ v.g./ml, 10-fold higher) to examine the temporal profile of damage in the same cortical region of S1BF (*Figure 6C*). At the checkpoints of 17-, 55-, 70-, and 92-day postinjection, neurons expressing GCaMP6m-$X_C$ were compared with GCaMP6m. Confocal microscopy with brain slices revealed that the percentage of infected neurons and the expression level of GCaMP6m-$X_C$ were close to their peaks on 17 days, suggesting that the ultrahigh dose could expedite GCaMP6m-$X_C$ expression to reach the high level. Most importantly, long-term, high-level expression of GCaMP6m-$X_C$ up to 92 days did not induce nuclear accumulation, whereas GCaMP6m at relatively lower concentration ($1 \times 10^{12}$ v.g./ml) already

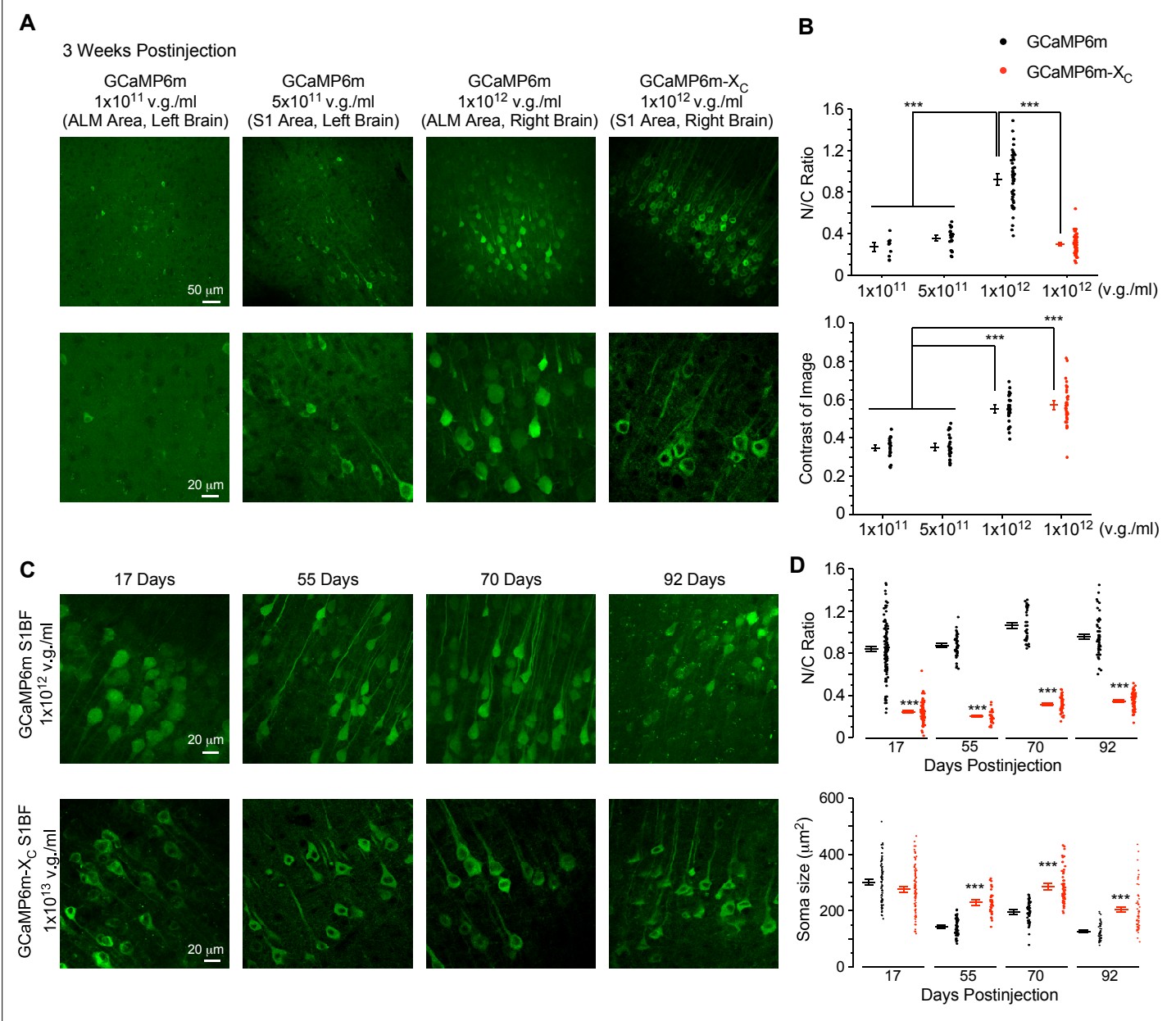

**Figure 6.** Evaluation of neuronal morphology for long-term in vivo expression of GCaMP-X versus GCaMP. (**A**) Confocal brain-slice images for ALM (anterolateral motor cortex) or S1 (primary somatosensory cortex). For the same mouse, AAV-*Syn*-GCaMP6m or AAV-*Syn*-GCaMP6m-$X_C$ (30 nl) of the indicated titers were injected into the left or right brain. Brain slices were dissected 3 weeks after virus injection. (**B**) N/C ratio (upper) and contrast of images (lower), summarized over three mice. (**C**) Confocal images of brain slices expressing GCaMP6m or GCaMP6m-$X_C$ acquired at different timepoints up to 92 days postinjection. AAV-*Syn*-GCaMP6m or AAV-*Syn*-GCaMP6m-$X_C$ (60 nl) viruses of the indicated titers were microinjected into the left or right S1BF (barrel field of S1) of the same mouse, respectively. (**D**) Summary of N/C ratio (upper) and soma size (lower) for the neurons expressing GCaMP6m or GCaMP6m-$X_C$ (eight mice in total). Standard error of the mean (SEM) and one-way analysis of variance (ANOVA) followed by Bonferroni for post hoc tests (criteria of significance: ***$p < 0.001$) were calculated when applicable.

The online version of this article includes the following figure supplement(s) for figure 6:

**Figure supplement 1.** Long-term effects of GCaMP-X with high-expression levels in vivo.

caused severe nuclear accumulation evidenced from 17 to 92 days (*Figure 6D*). Meanwhile, the soma size of GCaMP6m-infected neurons was significantly smaller than that of GCaMP6m-$X_C$ from 55 to 92 days, presumably due to impaired spontaneous $Ca^{2+}$ activities and related $Ca^{2+}$ signals in these neurons. To directly confirm the relative expression levels, immunocytochemistry was performed

on the brain slices infected by GCaMP6m and GCaMP6m-X$_C$ viruses, under similar conditions as in *Figure 6C, D*. Long-term (13 weeks) expression levels of AAV-*Syn*-GCaMP6m (at the concentrations of $5 \times 10^{11}$ and $1 \times 10^{12}$ v.g./ml) in the brains of adult mice were quantified by anti-GFP immuno-fluorescence (*Figure 6—figure supplement 1*). And both dosages resulted in significantly lower expression levels than AAV-*Syn*-GCaMP6m-X$_C$ of higher concentration ($1 \times 10^{13}$ v.g./ml), excluding the expression level as the cause of less damage. Of note, the soma size of neurons after long-term GCaMP-X expression was larger than GCaMP, while indistinguishable from the blank control neurons. In summary, in comparison with GCaMP, GCaMP-X exhibited high compatibility with neurons as desired by chronic Ca$^{2+}$ imaging.

## Transgenic GCaMP mice may have similar neuronal toxicities

Although the drawbacks of GCaMP were noticed at the very beginning of its invention and then improved by mechanism-based rational-design later on, GCaMP transgenic mice have been considered to be relatively safe in comparison with viral delivery of GCaMP. Nevertheless, recent studies reported that some transgenic mouse lines, such as Ai93 and Ai148, suffered from epileptiform activities (*Daigle et al., 2018*; *Steinmetz et al., 2017*). Based on our data thus far mostly by way of transient transfection and viral infection, we suspected that the mechanisms of side-effects are likely applicable to transgenic expression of GCaMP. Following up this hypothesis, brain slices and culture neurons from transgenic mice were examined from functional and morphological aspects.

Ai148 is a widely used transgenic line, for which TIGRE2.0 has been utilized for GCaMP6f to enhance its expression level, such that the damage by GCaMP is potentially more pronounced (*Daigle et al., 2018*). Using confocal fluorescence microscopy, we examined GCaMP-expressing neurons from the layer II–III cortex of the 6-month-old Rasgrf2-2A-dCre;Ai148 mice (with TMP-inducible expression of GCaMP6f) (*Figure 7A*). Nuclear accumulation of GCaMP was readily discernible, although it was relatively less severe than long-term expression of viral GCaMP6m of high doses (*Figure 6*). Next, dissected from newborn Rasgrf2-2A-dCre;Ai148 mice, cortical neurons were cultured, and subsequently 10 µM TMP was added to induce GCaMP expression. Similar to viral delivery, transgenic neurons were also subject to GCaMP6f perturbations, especially in nucleus-filled neurons. Neurite tracings indicated that the complexity and length of neurites were reduced in Ai148 neurons as compared to GFP control neurons from DIV 21 onwards (*Figure 7C*). The temporal profile of total neurite length indicated that neurite outgrowth was significantly slowed down or even halted on DIV 14, in comparison with the control neurons (GFP virus-infected and TMP treated) (*Figure 7D*). Consistently, N/C ratio of GCaMP indicative of nuclear accumulation was gradually increased along with the expression time up to 1 month (*Figure 7E*). Functionally, Ca$^{2+}$ waveforms of lower amplitude were acquired from Ai148 neurons expressing nucleus-filled GCaMP across the full month than the nucleus-excluded subgroup (*Figure 7F, G*), consistent with the previous results by GCaMP plasmids and viruses that the side-effects would be exacerbated by nuclear GCaMP. Similar results were obtained from analyzing the peak amplitude and integrated frequency of Ca$^{2+}$ oscillations by comparing nucleus-filled versus nucleus-excluded subgroups of neurons on DIV 14 or later (*Figure 7H, I*). Another trial of neurite tracing and Ca$^{2+}$ imaging with Ai148 neurons confirmed the effects and analyses described above (*Figure 7—figure supplement 1*).

Comparing with GFP control neurons from Ai140D mice, the potential effects of tTA (*Moullan et al., 2015*) were excluded from the major results and conclusions regarding GCaMP toxicities (*Figure 7—figure supplement 2*). Also, by adding TMP on DIV 14 to induce transgenic GCaMP6f expression at the mature stage, similar damage on neurite morphology and Ca$^{2+}$ oscillation was observed from Ai148 neurons, consistent with the previous notions that both premature and mature neurons are subject to GCaMP perturbations (*Figure 7—figure supplement 3*). In addition to chemical-inducible expression of GCaMP, newborn Emx1-Cre;Ai148 mice were deployed to constitutively express GCaMP6f (*Figure 7—figure supplement 4*). GCaMP6f started to accumulate in the nucleus at very early stage indexed by N/C ratio (criteria of 0.8). The damage was clearly evidenced when compared with the control neurons virally expressing GCaMP6m-X$_C$. Meanwhile, the major characteristics of spontaneous Ca$^{2+}$ oscillation in transgenic neurons were also significantly altered, resulting in relatively lower frequency, less synchronization, smaller amplitude, and abnormally wider FWHM.

In summary, the major findings by virus-infected neurons are applicable to transgenic mice, where GCaMP expression is either TMP inducible or constitutive. Morphological and functional analyses

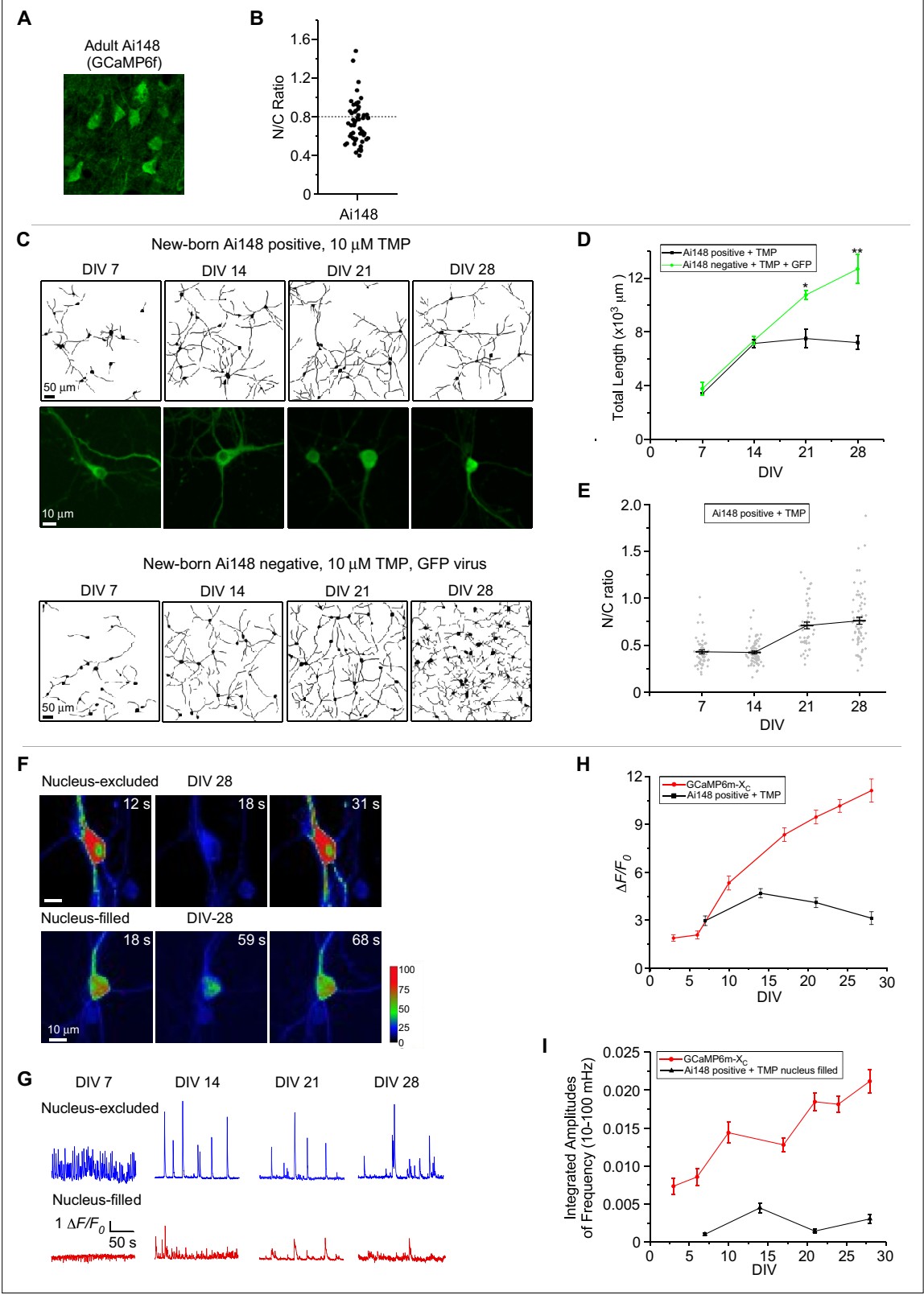

**Figure 7.** Chronic imaging for cultured cortical neurons of GCaMP transgenic mice. (**A**) Confocal brain-slice images of cortex layer II–III from 6-month-old GCaMP6f-positive transgenic mice (Rasgrf2-2A-dCre;Ai148 mice). GCaMP6f was examined ~4 months after induction of expression by intraperitoneal injection of TMP. (**B**) The N/C ratio of layer II–III neurons of adult Ai148 mice. (**C**) Neurite growth of cultured cortical neurons from newborn Ai148 mice. Neurite tracing for GCaMP-positive neurons (upper two rows) versus GCaMP-negative neurons (infected by GFP virus, bottom

*Figure 7 continued on next page*

*Figure 7 continued*

row), both added with 10 μM TMP to induce transgene expression. Zoomed confocal images illustrate subcellular distributions of GCaMP (middle row). Temporal profiles of total neurite length (**D**) or N/C ratio (**E**) for GCaMP-expressing Ai148 neurons, in comparison with GCaMP-negative GFP-infected neurons. (**F**) Time-laps images of single-neuron Ca$^{2+}$ dynamics on DIV 28 from the nucleus-excluded and nucleus-filled subgroups. Color coding indicates fluorescence intensity. (**G**) Spontaneous Ca$^{2+}$ activities of GCaMP-positive neurons in the nuclear-excluded and nuclear-filled subgroups from DIV 7 to DIV 28. Temporal profiles of the peak amplitudes ($\Delta F/F_0$, **H**) and the integrated amplitudes (over the frequency band of 10–100 mHz, **I**) for Ai148 neurons, compared with GCaMP-X$_C$. Data were based on 3 Ai148 mice (**A, B**), and two independent experiments from two independent culture preparations (**C–I**). Standard error of the mean (SEM) and two-tailed unpaired Student's *t*-test (**D**) (criteria of significance: *p < 0.05; **p < 0.01) were calculated when applicable.

The online version of this article includes the following figure supplement(s) for figure 7:

**Figure supplement 1.** Chronic evaluation of cultured cortical neurons from Ai148 mice with inducible GCaMP6f expression.

**Figure supplement 2.** Neuritogenesis of cortical neuron from transgenic mice with TMP-inducible GCaMP.

**Figure supplement 3.** Effects on cortical neurons by transgenic GCaMP6f expression induced at the mature stage.

**Figure supplement 4.** Long-term culturing and imaging of cortical neurons from Emx1-Cre;Ai148 mice with constitutive GCaMP6f expression.

strongly suggest that cortical neurons in transgenic GCaMP6 mice are also subject to GCaMP toxicity similar to virus-infected neurons.

## Discussion

In this work, we applied GCaMP-X with reduced cell toxicity or enhanced neuron compatibility, to monitor Ca$^{2+}$ dynamics across multiple days/weeks both in vitro and in vivo. By way of transient transfection, viral infection or transgenic expression, GCaMP of prolonged/excessive expression caused neuronal toxicities presumably due to its perturbations on endogenous apoCaM interactions, which were significantly reduced by rationally designed GCaMP-X. By relieving the concerns on the time and level of probe expression, GCaMP-X provides a simple solution for chronic calcium imaging in alternative to circumventing GCaMP toxicity. GCaMP-X paves the way to unexplored directions previously impeded or discouraged due to GCaMP perturbations.

### Available design solutions to avoid the side-effects of CaM-based GECI

To better utilize CaM-based GECIs in vitro and in vivo, solutions with no or minimum side-effects are in need. In utero electroporation and viral infection often result in high-expression levels particularly near the injection site, and some lines of transgenic mice using weaker promoters could control probe expression within the low levels to alleviate nuclear accumulation (*Akerboom et al., 2012*; *Dana et al., 2019*). Since probes present in the cytosol are able to bind apoCaM targets including Ca$_V$1 channels (*Yang et al., 2018*), neurons may still suffer from the toxicity of the probes. Evidently, cytosolic GCaMP affected neural excitability in transgenic mice expressing GCaMP5G or GCaMP6 (*Steinmetz et al., 2017*). To overcome these problems, one solution is to substitute the core components of GECI design, for example, to utilize troponin C from muscle as a Ca$^{2+}$-binding motif (*Mank et al., 2008*). The TN-XXL has been claimed to be suitable for chronic imaging potentially benefitted from its design basis (less likely to bind endogenous proteins in neurons). However, the TN-XXL solution has at least two shortcomings. First, TNXXL is a FRET-based ratiometric sensor, of which the dynamic range is limited by FRET methods and indeed much narrower than GCaMP. Second, the Ca$^{2+}$-binding motif from mammalian troponin C has the canonical EF-hands (resembling CaM), thus still possible to perturb neurons by binding endogenous targets at the apo states, which needs further investigations.

The approach adopted by this work is to introduce an additional protective motif that specifically binds apoCaM within the probe (*Figure 1A*). Such CBM has been fused onto the N-terminus of conventional GCaMPs (from GCaMP3 to jGCaMP7) to construct a new series of GCaMP-X correspondingly. When Ca$^{2+}$ level is low, CBM successfully prevents apoCaM contained within GCaMP-X from interfering with Ca$_V$1 channels and other important apoCaM targets (*Figure 1—figure supplement 3*). Once Ca$^{2+}$ concentration rises, M13 binds to Ca$^{2+}$/CaM with high affinity, without altering Ca$^{2+}$-sensing characteristics of GCaMP-X inherited from years of efforts and improvements. With GCaMP-X as the proof-of-principle, the design rules centered with apoCaM/Ca$^{2+}$-CaM binding are potentially applicable to CaM-based sensors or actuators of a broader scope (*Grødem et al.,*

*2021*; *Haiech et al., 2019*), which may face similar challenges or problems to those associated with enhanced/prolonged expression of GCaMP.

## Spontaneous Ca²⁺ activities in association with neural development and degeneration

While the membrane voltage is oscillating, cellular $Ca^{2+}$ signals are also fluctuating, closely involved in neuronal development and circuit formation both in intro and in vivo (*Kirkby et al., 2013*; *Sun et al., 2010*). Meanwhile, in line with the aforementioned calcium hypothesis, dysregulated spontaneous $Ca^{2+}$ activities would lead to defective morphology and functions of neurons, and eventually neural diseases (*Harr and Distelhorst, 2010*; *Khan et al., 2020*; *Nicotera and Orrenius, 1998*). In $Ca^{2+}$ imaging experiments, $Ca^{2+}$ fluorescence signals and electrical activities are often referred to each other since action potentials initiated by $Na^+$ channels would subsequently drive the fast opening of $Ca^{2+}$ channels that mediate $Ca^{2+}$ influx. Electrical recording via MEA (multiple electrode array) has been widely applied in long-term brain/neuron monitoring (*Obien and Frey, 2019*; *Shafer, 2019*), among other methods. On the other hand, both membrane potentials and ion fluxes ($Na^+$ or $Ca^{2+}$) could have sophisticated mechanisms and specific consequences, for example, $Ca^{2+}$ oscillations of different forms: subthreshold oscillations by L-type $Ca^{2+}$ channels, or intracellular $Ca^{2+}$ fluctuations by intracellular $Ca^{2+}$-release channels (*Chan et al., 2007*; *Uhlén and Fritz, 2010*). GECIs are promising tools to overcome the limitations of many other methods including MEA, if the cell compatibility issues could be resolved as demonstrated by GCaMP-X in this work. GECI imaging methods directly and faithfully capture $Ca^{2+}$ activities at different loci in the brain or within the cell, allowing high spatial–temporal resolution of concurrent morphological/functional imaging. Subcellular $Ca^{2+}$ oscillations may be responsible for different aspects of neurogenesis and neuritogenesis, awaiting future investigations with organelle-specific GCaMP-X, such as the nuclear version GCaMP-X$_N$. Earlier $Ca^{2+}$ activities (weak fluctuations of higher frequency) may represent spontaneous activity before synapse or network formation (*Spitzer, 2006*). At the later stage, synchronized $Ca^{2+}$ oscillations (of lower frequency) emerge, along with dramatic changes in morphology and other aspects of development. Autonomous oscillations of SNc neurons are tightly coupled with $Ca_V1.3$ channels which may underpin Parkinson's disease or neural aging (*Chan et al., 2007*; *Guzman et al., 2009*). Cultured cortical slices and hiPSC-derived cortical neurons also suggest that L-type $Ca^{2+}$ channels are crucial for both spontaneous $Ca^{2+}$ activities and neuronal development (*Horigane et al., 2021*; *Plumbly et al., 2019*). A similar mechanism is likely shared by the correlation between spontaneous/oscillatory $Ca^{2+}$ activities and neuronal morphology/development unveiled in this study. Expression, trafficking and functions of ion channels and membrane receptors are also subject to regulations by activities of different patterns (*Ruffinatti et al., 2013*; *Spitzer, 2006*; *Toth et al., 2016*), and chronic GECI imaging is expected to help elucidate these compound effects and mechanisms. In this work, we have particularly focused on spontaneous $Ca^{2+}$ activities of cortical neurons in association with neuronal morphology both in vitro (*Figures 3 and 4*) and in vivo (*Figures 2, 5 and 6*), and in both neonatal (*Figures 3 and 4*) and mature neurons (*Figure 5—figure supplement 1* and *Figure 7—figure supplement 3*), as the exemplars to demonstrate the performance of chronic GCaMP-X imaging. Importantly, since such $Ca^{2+}$ activity-neuronal morphology coupling was perturbed by GCaMP under various testing conditions, we are expecting a broad scope of applications awaiting GCaMP-X to explore both in vitro and in vivo.

## Improved neuron compatibility of GCaMP-X

GCaMP, as widely applied GECI, has been evolved into its eighth version (*Grødem et al., 2021*), with enhanced sensitivity, brightness and kinetic properties tailored to specific imaging purposes, including single action potentials and activities in neuronal populations and microcompartments (*Dana et al., 2019*). However, the cytotoxicity of GCaMP has been a persistent problem from early on, mainly due to the fact that the CaM-centered schemes of GCaMP have been largely inherited across generations of its design (*Akerboom et al., 2012*; *Chen et al., 2013*; *Dana et al., 2019*; *Nakai et al., 2001*; *Tallini et al., 2006*; *Tian et al., 2009*). In vivo $Ca^{2+}$ imaging with GCaMP viruses is facing the dilemma of safety versus reliability. On the one hand, reducing GCaMP levels could alleviate or postpone the cytotoxicity, but low levels of expression would also reduce the contrast

and SNR (*Rose et al., 2014*). On the other hand, increasing the expression level of GCaMP would enhance the data quality of $Ca^{2+}$ imaging, but exacerbate the damage to neurons (*Resendez et al., 2016*). It is not surprising that GCaMP transgenic mice have encountered with similar problems. Several lines of GCaMP transgenic mice reported earlier, such as Emx1-Cre;Ai38 GCaMP3 transgenic mice, attempted to resolve the safety issue by restricting the expression to ultra-low levels (~5 μM) (*Zariwala et al., 2012*), but sacrificing the imaging quality (*Rose et al., 2014*). The mouse lines reported recently, such as Emx1-Cre;CaMK2α-tTA;Ai93 GCaMP6f transgenic mice and Slc17a7-IRES2-Cre;Ai148 GCaMP6f transgenic mice (*Daigle et al., 2018*; *Madisen et al., 2015*), managed to elevate the expression levels. However, epileptiform activities have been observed from these mice presumably due to GCaMP perturbations (*Daigle et al., 2018*; *Steinmetz et al., 2017*). One bypass solution is to conditionally induce GCaMP expression to conduct GCaMP imaging within a time window, which would be much less feasible for long-term expression and/or chronic imaging.

Instead, GCaMP-X allows long-term and high-level expression to increase the quality and reliability of $Ca^{2+}$ imaging while reducing neuronal toxicities. For in vitro studies, investigations of long-term $Ca^{2+}$ dynamics are largely hindered by the cytotoxicity of GECIs or dyes (*Rose et al., 2014*; *Smith et al., 2018*). GCaMP-X is well suited for longitudinal $Ca^{2+}$ dynamics, for example, during neural development as in this study, for high-throughput screening of long-term drug effects (*Vetter et al., 2020*), or in other similar scenarios. For in vivo studies, due to the concerns known to GCaMP, false-negative or false-positive results by nuclear-filled GCaMP or under-expressed GCaMP are misleading especially in imaging large populations of neurons (*Resendez et al., 2016*). The existing reports have not yet reached an agreement regarding whether nuclear GCaMP would cause neuronal toxicities with a significant impact during the early postinjection phase, for example, *Figures 2 and 5*. Such discrepancy may reflect different doses of virus injection, different expression levels of GCaMP or even different types of neurons in different brain regions. Alternatively, it may still reinforce that neuronal damage could still exist even within OTW, as a precaution for selecting neurons and planning experiments. In this regard, GCaMP-X provides a viable option with higher SNR, more healthy neurons, lower neurotoxicity, prolonged expression/imaging, and meanwhile less experimental complexity.

Additional control experiments would help evaluate how close GCaMP-X data are to the reality, considering potential $Ca^{2+}$-buffering effect intrinsic to $Ca^{2+}$ probes and also other factors. Applicable controls were incorporated to better evaluate GCaMP-X data, for example, Ai140D mice (GFP, *Figure 7—figure supplement 2*) and Fluo-4 AM ($Ca^{2+}$ dye, *Figure 3—figure supplement 4*). The results have been encouraging in that GCaMP-X neurons were nearly indistinguishable in the morphological and functional aspects from Ai140D neurons expressing GFP or loaded with Fluo-4 AM. The feedbacks from GCaMP-X applications should continue to help clarify this matter in the future.

## Materials and methods

### Key resources table

| Reagent type (species) or resource | Designation | Source or reference | Identifiers | Additional information |
|---|---|---|---|---|
| Strain, strain background (*Mus musculus*) | ICR | Institute of Cancer Research | | Produced by Tsinghua University & Beihang University |
| Strain, strain background (*Mus musculus*) | C57 | Jackson Labs | C57BL/6J; stock no. 000664 | Produced by Tsinghua University |
| Strain, strain background (*Mus musculus*) | Ai148 (or Ai148D) | Jackson Labs | B6.Cg-Igs7tm148.1(tetO-GCaMP6f,CAG-tTA2)Hze/J; stock no.030328 | |
| Strain, strain background (*Mus musculus*) | Rasgrf2-2A-dCre | Jackson Labs | B6;129S-Rasgrf2tm1(cre/folA)Hze/J; stock no.022864 | |

*Continued on next page*

*Continued*

| Reagent type (species) or resource | Designation | Source or reference | Identifiers | Additional information |
|---|---|---|---|---|
| Strain, strain background (*Mus musculus*) | Ai140D | Jackson Labs | B6.Cg-Igs7tm140.1(tetO-EGFP,CAG-tTA2)Hze/J; stock no.030220 | |
| Strain, strain background (*Mus musculus*) | Emx1-cre | Jackson Labs | B6.129S2-Emx1tm1(cre)Krj/J; stock no.005628 | |
| Cell line (*Homo sapiens*) | Kidney (normal epithelial, embryo) | ATCC | HEK293 | |
| Antibody | anti-GFP antibody (Rabbit polyclonal) | Abcam | Cat#ab290; RRID: AB_2313768 | WB (1:1000); IF (1:200) |
| Antibody | anti-GAPDH antibody (Rabbit polyclonal) | Gene-Protein Link | Cat#P01L081 | WB (1:10,000) |
| Antibody | anti-Histone H3 antibody (Mouse monoclonal) | Beyotime | Cat#AF0009; RRID: AB_2715593 | WB (1:1000) |
| Antibody | anti-Calmodulin 1/2/3 antibody (Rabbit monoclonal) | Abcam | Cat#ab45689; RRID: AB_725815 | WB (1:10,000) |
| Antibody | anti-His antibody (Rabbit monoclonal) | Gene-Protein Link | Cat#P01L071 | WB (1:2000) |
| Antibody | HRP-conjugated Affinipure Goat Anti-Mouse IgG (H + L) | Proteintech | Cat#SA00001-1; RRID: AB_2722565 | WB (1:10,000) |
| Antibody | HRP-conjugated Affinipure Goat Anti-Rabbit IgG (H + L) | Proteintech | Cat#SA00001-2; RRID: AB_2722564 | WB (1:10,000) |
| Antibody | anti-NeuN antibody (Mouse monoclonal) | Millipore | Cat#MAB377; RRID: AB_2298772 | IF (1:500) |
| Antibody | Goat anti-Mouse IgG (H+L) Alexa Fluor 568 (Goat polyclonal) | Invitrogen | Cat#A11004; RRID:AB_2534072 | IF (1:800) |
| Antibody | Goat anti-Rabbit IgG (H+L) Alexa Fluor 647 (Goat polyclonal) | Invitrogen | Cat#A21244; RRID:AB_2535812 | IF (1:800) |
| Chemical compound, drug | DAPI | Beyotime | 2-(4-Amidinophenyl)-6-indolecarbamidine dihydrochloride; Cat#C1002 | IF (1:1000) |
| Chemical compound, drug | TMP | Sigma | Trimethoprim, Cat#T7883-5G | Dilute to DMSO |
| Chemical compound, drug | Fluo-4 AM | Beyotime | Cat#S1060 | |
| Chemical compound, drug | Pluronic F-127 | Beyotime | Cat#ST501-1g | |
| Commercial assay or kit | Nuclear extraction kit | Abcam | Cat#113474 | |
| Software, algorithm | GraphPad Prism v8.0.1 | GraphPad Software | RRID:SCR_002798 | |
| Software, algorithm | Imaris Viewer x64 v7.7.2 | Oxford Instruments Group | RRID:SCR_007370 | |
| Software, algorithm | Fiji v6.6.1 | National Institutes of Health | RRID:SCR_002285 | |
| Software, algorithm | Origin 2019b | OriginLab | RRID:SCR_014212 | |

*Continued*

| Reagent type (species) or resource | Designation | Source or reference | Identifiers | Additional information |
|---|---|---|---|---|
| Software, algorithm | Clampex | Molecular Devices | RRID:SCR_011323 | |
| Software, algorithm | Matlab | MathWorks | RRID:SCR_001622 | |
| Recombinant DNA reagent | pGP-CMV-GCaMP6m (plasmid) | PMID:23868258 | RRID:Addgene_40754 | |
| Recombinant DNA reagent | pEGFP-N1-GCaMP6m-$X_C$ (plasmid) | PMID:29666364 | RRID:Addgene_111543 | |
| Recombinant DNA reagent | pGP-CMV-jGCaMP7b (plasmid) | PMID:31209382 | RRID:Addgene_104484 | |
| Recombinant DNA reagent | pEGFP-N1-jGCaMP7b-$X_C$ (plasmid) | Created in this study | RRID:Addgene_178361 | NES tag added to jGCaMP7b-X, generated by Liu Lab. |
| Recombinant DNA reagent | pEGFP-N1-jGCaMP7b-$X_N$ (plasmid) | Created in this study | RRID:Addgene_178362 | NLS tag added to jGCaMP7b-X, generated by Liu Lab. |
| Recombinant DNA reagent | pcDNA3-$\alpha_{1DL}$ (plasmid) | PMID:35589958 | | Plasmid to express full length $\alpha_{1DL}$, generated by Liu Lab. |
| Recombinant DNA reagent | pcDNA3-$\alpha_{1DL}$-3xFlag (plasmid) | PMID:35589958 | | Plasmid to express full length $\alpha_{1DL}$ with C-terminal 3xFlag tag, generated by Liu Lab. |
| Recombinant DNA reagent | pcDNA3-Flag-CaMBD_$\alpha_{1DL}$ (plasmid) | Created in this study | | Plasmid to express CaMBD of $\alpha_{1DL}$ with N-terminal Flag tag, generated by Liu Lab. |
| Recombinant DNA reagent | pcDNA3.1-YFP-Ng_S36A-Myc-HisA (plasmid) | Created in this study | | Plasmid to express Ng_S36A with C-terminal Myc-His tag, generated by Liu Lab. |
| Other | AAV2/DJ-*Syn*-GCaMP6m | Hanbio Biotechnology | | Adeno-associated virus to express GCaMP6m in vitro |
| Other | AAV2/DJ-*Syn*-GCaMP6m-$X_C$ | Hanbio Biotechnology | | Adeno-associated virus to express GCaMP6m-$X_C$ in vitro |
| Other | AAV2/DJ-*Syn*-jGCaMP7b | Hanbio Biotechnology | | Adeno-associated virus to express jGCaMP7b in vitro |
| Other | AAV2/DJ-*Syn*-jGCaMP7b-$X_C$ | Hanbio Biotechnology | | Adeno-associated virus to express jGCaMP7b-$X_C$ in vitro |
| Other | AAV2/DJ-*Syn*-ZsGreen | Hanbio Biotechnology | | Adeno-associated virus to express ZsGreen in vitro |
| Other | pLenti-*Syn*-mCherry | OBiO Technology | | pLenti virus to express mCherry in vitro |
| Other | AAV2/9-*Syn*-GCaMP6m | BrainVTA | | Adeno-associated virus to express GCaMP6m in vivo |
| Other | AAV2/9-*Syn*-GCaMP6m-$X_C$ | BrainVTA | | Adeno-associated virus to express GCaMP6m-$X_C$ in vivo |

## Molecular biology

The plasmids of jGCaMP7b-$X_C$ and jGCaMP7b-$X_N$ were constructed by replacing previously reported GCaMP6m-$X_C$ or GCaMP6m-$X_N$ (*Yang et al., 2018*) with appropriate PCR-amplified segments from jGCaMP7b via unique EcoRI and HindIII sites, or EcoRI and NotI sites, respectively. pcDNA3-Flag-CaMBD_$\alpha_{1DL}$ was generated by replacing YFP from pcDNA3-YFP-preIQ$_3$-IQ$_D$-PCRD$_D$ (*Yang et al., 2022*), with PCR-amplified Flag (DYKDDDDK) to by KpnI and NotI. pcDNA3.1-YFP-Ng_S36A-Myc-His was generated by inserting the PCR-amplified segments of neurogranin (NM_024140.2) containing S36A into a customized pcDNA3.1-MCS-Myc-His vector via unique EcoRI and HindIII sites.

## Mice

Procedures involving animals have been approved by local institutional ethical committees (IACUC in Tsinghua University and Beihang University), similar to the previous protocol (*Huber et al., 2012*). In

vivo experiments were conducted with adult mice (C57BL/6J, both male and female) of 2–6 months old. In total, 23 mice (C57BL/6J) were used for expression tests and functional tests (GCaMP6m and GCaMP6m-X$_C$). Three mice of Ai148;Rasgrf2-2A-dCre (Jax #030328; Jax #022864) were used for brain-slice imaging to examine transgenic GCaM6f neurons. Expression of GCaMP6f in Ai148;Rasgrf2-2A-dCre mice was induced with antibiotic Trimethoprim (TMP) by intraperitoneal injection with the dose of 0.25–0.5 mg/g in vivo (Sando et al., 2013). In vitro experiments were based on data from 82 mice (P0–P1, both male and female). Fifty-two ICR mice were used for expression and functional tests of Ca$^{2+}$ indicators. Two newborn Ai148;Emx1-Cre mice were used to persistently express GCaMP in cultured neurons. In the tests of cortical neurons expressing GCaMP6f from Ai148;Rasgrf2-2A-dCre mice, 15 Ai148; Rasgrf2-2A-dCre GCaMP6f-positive mice were compared with 7 ICR control mice, 3 Ai140D;Rasgrf2-2A-dCre GFP-positive mice and 2 Ai140D-positive;Ragrf2-2A-dCre GFP-negative mice. Expression of GCaMP6f or GFP was induced by directly adding 10 μM TMP into growth medium of cultured neurons after dissection.

## Dissection and culturing of cortical neurons

Cortical neurons were dissected from newborn mice. Isolated tissues of cortex were digested with 0.25% trypsin for 15 min at 37°C. Then digestion was terminated by Dulbecco's modified Eagle medium (DMEM) supplemented with 10% fetal bovine serum (FBS). The cell suspension was sieved through a filter and centrifuged at 1000 rpm for 5 min. The cell pellet was resuspended in DMEM supplemented with 10% FBS and were plated on poly-D-lysine-coated 35 mm No. 0 confocal dishes (In Vitro Scientific). After 4–6 hr, neurons were maintained in Neurobasal medium supplemented with 2% B27 and 1% glutaMAX-I (growth medium), and cultured in the incubator with temperature of 37°C and 5% CO$_2$. Fresh growth medium was supplemented to neurons every 3–4 days to maintain the volume of 2 ml growth medium. All animals were obtained from the laboratory animal research center, Tsinghua University. Procedures involving animals has been approved by local institutional ethical committees (IACUC in Tsinghua University and Beihang University).

## Virus infection on cultured neurons

All viruses for infection of cultured neurons were provided by Hanbio Biotechnology, China. The neuron broad-spectrum promoter *Syn* and AAV2/DJ serotypes were selected for neuro-specific expression of GFP, GCaMP or GCaMP-X in cultured cortical neurons. 1 μl 1 × 10$^{12}$ v.g./ml of the desired kinds of viruses were added to growth medium on DIV 0 unless otherwise noted. The same batches of cortical neurons were simultaneously observed for comparison. The expression of GCaMP and GCaMP-X was detectable on DIV 3, reached the peak on DIV 7 and sustained the high level up to 1 month. All experiments in vitro were repeated independently at least twice.

## Ca$^{2+}$ imaging with GCaMP or GCaMP-X in cultured cortical neurons

Ca$^{2+}$ imaging of neurons expressing GCaMP or GCaMP-X was acquired by confocal microscopy (A1RMP, Nikon, Japan; Dragonfly 200, Andor, England). 488 nm laser was used for excitation. 35 mm confocal dish containing cultured cortical neurons was set in the live-cell imaging culture chamber of the confocal microscope to maintain the environment of 37°C, 5% CO$_2$ and ~95% humidity. Sampling rate of images was at 1–5 Hz and 3–5 view fields were selected from each dish. Fluorescence intensity (*F*) was subtracted from its background. $F_0$ is the baseline fluorescence averaged from five data points at rest, and $\Delta F = F - F_0$. $\Delta F/F_0$ serves as the index for Ca$^{2+}$ dynamics. Ca$^{2+}$ waveforms were analyzed by the gadget of Quick Peaks in Origin software with the Three-Standard-Deviations Rule (values >3 SD). On and off rates were characterized by the time to rise up or decay down to 50% of the maximum ($\Delta F/F_0$), respectively. And the FWHM is defined as the duration of time between the (upward and downward) half-maximum timepoints. The mean of correlation coefficients based on Spearman Rank Correlation Coefficient in Origin software (Schaworonkow and Nikulin, 2019) was applied to quantify the degree of neuronal correlations based on all the traces of spontaneous Ca$^{2+}$ signals per view (Sumi et al., 2020). FFT analyses were performed in Origin software. One-sided is selected for spectrum type and amplitude is define as below:

$$\text{FFT Amplitude} = 2\sqrt{Re^2 + Im^2}/n$$

Here, *Re* and *Im* are the real and imaginary parts of FFT results, and *n* is the size of the input signal.

Neurite tracings were depicted with Imaris 7.7.2 from the images under average intensity projection. Analyses on neuronal morphology and $Ca^{2+}$ oscillations were performed with at least 30 neurons at each timepoint in each independent experiment. Fluorescence intensities of $Ca^{2+}$ dynamics in neurons were color coded by Matlab (Mathworks) and Fiji.

## Transfection, confocal fluorescence imaging, and analysis of neurite morphology

2 µg of cDNA encoding jGCaMP7b, jGCaMP7b-$X_C$, or jGCaMP7b-$X_N$ and 1 µg cDNA encoding CFP (for labeling the soma area and neurites) were transiently transfected into DIV 5–7 cultured cortical neurons by Lipofectamine 2000 (Invitrogen) with a typical protocol according to the manual. The opti-MEM containing plasmids and Lipofectamine 2000 was added to the Neurobasal medium for transfection. After 2 hr, neurons were maintained in Neurobasal medium supplemented with 2% B27, 1% glutaMAX-I for at least 2 days before analyzing neurite morphology.

Fluorescence imaging of cultured cortical neurons was performed on ZEISS Laser Scanning Confocal Microscope (LSM710, Carl Zeiss) and ZEN 2009 software. N/C ratio of GCaMP or GCaMP-X was calculated by the ratio of fluorescence intensity (nuclear/cytosolic). Measurement of the total length and *Sholl* analysis for neurites were performed with Imaris 7.7.2 (Bitplane). Only non-overlapping neurons were selected for analysis and images of at least 24 neurons from two independent culture preparations were analyzed. Neurite tracings were depicted with Imaris 7.7.2 in CFP channel.

## Craniotomy and in vivo virus injection

Wildtype mice were used for virus injection and craniotomy under isoflurane anesthesia (5% for induction, 1–1.5% during surgery). AAV2/9-*Syn*-GCaMP6m-$X_C$ (1.0 × $10^{13}$ v.g./ml, customized by BrainVTA, Wuhan, China) virus was tested in the primary somatosensory cortex (S1BF: AP −1.5, ML −3.0, DV 0.2/0.4, in mm), in comparison with AAV2/9-*Syn*-GCaMP6m (1.0 × $10^{12}$ v.g./ml, BrainVTA, Wuhan, China) virus as a control.

Craniotomy was done 3 weeks after virus injection. A piece of skull above S1BF was removed to expose a square imaging window (~3 × 3 mm, centered on S1BF) and the cortex was protected by a hand-cut glass window using #1 coverglass. Then the glass window was fixed using adhesive (Krazy glue, Elmer's Products Inc) and dental cement. A head-post was also fixed to the posterior area of the mouse head using dental cement. 0.2 ml flunixin meglumine (0.25 mg/ml) was subcutaneously injected after surgery for 3 consecutive days.

## Preparations of brain slices and image analysis

Mice were anaesthetized by intraperitoneal injection of avertin solution (250–500 mg/kg). Then they were transcardially perfused with phosphate-buffered saline (PBS) followed by 4% PFA (paraformaldehyde) solution. Brains were immersed in 4% PFA solution overnight and were embedded in 2.5% agarose gel for slicing operation. Slices were obtained using the Lecia vibratome (LeciaVT1200S) with proper parameters including depth, speed and thickness of the brain section (50, 70, or 100 µm). Contrast of image was estimated by the equation below:

$$\text{Contrast of image} = 1 - \frac{2}{(F/F_{\text{background}} + 1)}$$

## In vivo two-photon $Ca^{2+}$ imaging

A two-photon random access mesoscope controlled with ScanImage 2017 (Vidrio Technologies) was used for in vivo $Ca^{2+}$ imaging (*Pologruto et al., 2003*; *Sofroniew et al., 2016*). Images (512 × 512 pixels, 600 × 600, or 300 × 300 µm²) of L2/3 cells (150–250 µm under pia) in S1BF were collected at 7.4 Hz frame rate. Laser power (970 nm) was up to 60 mW out of objective. Calcium signal was extracted using CaImAn toolbox and data analysis was performed using Matlab (*Giovannucci et al., 2019*).

For functional test, imaging was carried out together with contralateral whisker stimulation using a 1.2-mm-diameter pole (~3 mm in amplitude, 10 vibrations in 0.5 s or 1 s for each trial). For each ROI, 20–40 trials were performed and calcium signaling was aligned to the onset of the whisker stimulation.

The fluorescence of each neuron was measured by averaging all pixels within the ROI (regions of interest) and corrected for neuropil contamination. The fluorescence signal was estimated by the equation below:

$$F_{cell}\left(t\right) = F_{measured}\left(t\right) - r * F_{neuropil}\left(t\right)$$

where $r = 0.7$ and $F_{neuropil}(t)$ was measured by averaging the fluorescence signal of all pixels within a 40 μm radium from the cell (*Chen et al., 2013*).

Signal-to-noise (SNR) was calculated as the ratio of $F_{max}/F_0$ to standard deviation of the filtered trace, 1 s period right before the whisker stimulate.

## Whole-cell electrophysiology

HEK293 cells (ATCC) were cultured in 60 mm dishes and checked by PCR with primers 5′-GGCGAATG GGTGAGTAACACG-3′ and 5′- CGGATAACGCTTGCGACCTATG-3′ to ensure free of mycoplasma contamination. Recombinant channels by $\alpha_{1DL}$, $\beta_{2a}$ (M80545), and $\alpha_2\delta$ (NM012919.2) subunits (4 μg of cDNA for each) were transiently transfected according to established calcium phosphate protocol (*Liu et al., 2017a*; *Liu et al., 2017b*; *Liu et al., 2010*). To enhance expression, cDNA for simian virus 40T antigen (1 μg) was also cotransfected. Additional 2 μg of cDNA of jGCaMP7b or jGCaMP7b-$X_C$ was added as required in cotransfections. Whole-cell recordings of transfected HEK293 cells were performed at room temperature (25°C) using an Axopatch 200B amplifier (Axon Instruments). The internal solutions contained (in mM): CsMeSO$_3$, 135; CsCl, 5; MgCl$_2$, 1; MgATP, 4; HEPES, 5; and EGTA, 5; at 290 mOsm adjusted with glucose and at pH 7.3 adjusted with CsOH. The extracellular solution contained (in mM): TEA-MeSO$_3$, 135; HEPES, 10; CaCl$_2$ or BaCl$_2$, 10; 300 mOsm, adjusted with glucose and at pH 7.3 adjusted with TEAOH. Whole-cell currents were generated from a family of step depolarizations (−70 to +50 mV from a holding potential of −70 mV and step of 10 mV).

## Western blot

HEK293 cells or cortical neurons were washed by PBS three times, followed by being lysed in lysis buffer RIPA (radio immunoprecipitation assay) with protease inhibitor cocktail (Beyotime, P1006) for 20 min and centrifuged for 5 min at 14,000 × $g$ at 4°C. Loading buffer was added to the supernatant. Then the sample were boiled for 7 min. Proteins were separated using 10% sodium dodecyl sulfate–polyacrylamide gel electrophoresis and transferred to a PVDF (polyvinylidene fluoride) membrane for 90 min. Then PVDF membrane was blocked in 5% non-fat dry milk and incubated with primary anti-bodies overnight at 4°C. Next, the PVDF membrane was washed three times with 1× TBST at room temperature with shaking, and incubated with secondary antibodies for 1–2 hr at room temperature then washed with 1× TBST for three times again. The membrane was coved with ECL chemilumi-nescent liquid (beyotime, P0018FM) before detection with an enhanced chemiluminescence system. Three or more independent replicates were performed for each experiment. Cytoplasmic proteins were extracted using nuclear extraction kit (Abcam, ab113474) following the instructions. Cytoplasmic proteins were collected by removing the nuclear proteins extracted via the kit.

## Coimmunoprecipitation assay

HEK293 cells were transfected by Lipofectamine and cultured for 2 days before cell lysates were prepared by lysis buffer RIPA (with protease inhibitor cocktail, Beyotime, P1006) and centrifugation at 14,000 × $g$ for 5 min at 4°C. The supernatants were subjected to coimmunoprecipitation by using 20 μl anti-Flag or anti-Myc Magnetic Beads (Cat# B26102, B26301, bimake); and 5 mM EGTA overnight at 4°C. Beads were washed with PBST (1xPBS+0.5% Tween 20) for three times. Proteins were separated by sample loading buffer and boiled for 7 min. Then western blot was performed using the antibodies as indicated. Three or more independent replicates were performed for each experiment.

## Immunofluorescence staining

For immunostaining of cultured cortical neurons: Cortical neurons were fixed with PBS + 4% PFA for 15 min at room temperature, and washed three times by PBS. Fixed neurons were permeabilized in PBS + 0.3% Triton X-100 for 10 min, blocked in the 10% goat serum in PBS for 60 min, and then incubated in primary antibodies + 10% goat serum + PBS overnight in 4°C. Next day, cells were washed

three times by PBS with gentle shaking, incubated in PBS + secondary antibodies for 60 min, and then washed three times by PBS with gentle shaking.

For immunostaining of brain slices: Sections after slicing (50 μm thickness) were immersed in 0.3% PBST solution for 15 min and the solution was renewed every 5 min. Then, sections were blocked in blocking solution (0.3% PBST solution with 3% bovine serum albumin) for 1 hr at room temperature and stained with primary antibodies at 4°C for 36 hr. After washing three times by PBS, sections were incubated with secondary antibodies in turn for 60 min. Finally, sections were rinsed three times by PBS and prepared for visualization.

## Fluorescence Ca$^{2+}$ imaging with Fluo-4 AM

Neurons were loaded with 2 μM Fluo-4 AM and 0.05% (wt/vol) Pluronic F127 in neurobasal medium at 37°C for 20 min in dark. Then neurons were gently washed twice with preheated 1× PBS. Neurons were incubated with neurobasal medium for 10 min in dark. Fluo-4 AM was excited at 488 nm, and emission signals were detected in 521 nm. Images were obtained using a ×20 objective with 512 × 512 pixels.

## Data analysis and statistics

Data were analyzed in Matlab, OriginPro, and GraphPad software. Standard error of the mean and two-tailed Student's *t*-test or one-way analysis of variance followed by Bonferroni for post hoc tests were calculated when applicable. Criteria of significance: *$p < 0.05$; **$p < 0.01$; ***$p < 0.001$; and *n.s.* denotes 'not significant'. All experiments were performed at least twice with appropriate sample sizes. Analyses of data were individually performed by at least three persons. Key experiments, such as the in vivo tests, were performed by one person and analyzed by other persons to avoid the potential bias.

## Acknowledgements

We thank all X-Lab members for discussions and help. This work is supported by grants from Natural Science Foundation of China (81971728, 3217099, 11902021, 11827803 and U20A20390) and of Beijing (7191006 and 5204037), and open fund of the Key Laboratory for Biomedical Engineering of Ministry of Education, Zhejiang University.

## Additional information

### Funding

| Funder | Grant reference number | Author |
|---|---|---|
| National Natural Science Foundation of China | 81971728 | Xiaodong Liu |
| Natural Science Foundation of Beijing Municipality | 7191006 | Xiaodong Liu |
| Zhejiang University | Open Fund of the Key Laboratory for Biomedical Engineering of Ministry of Education | Xiaodong Liu |
| National Natural Science Foundation of China | 32170998 | Zengcai V Guo |
| National Natural Science Foundation of China | 11902021 | Yaxiong Yang |
| Natural Science Foundation of Beijing | 5204037 | Yaxiong Yang |
| National Natural Science Foundation of China | U20A20390 | Yubo Fan |

| Funder | Grant reference number | Author |
|---|---|---|
| National Natural Science Foundation of China | 11827803 | Yubo Fan |

The funders had no role in study design, data collection, and interpretation, or the decision to submit the work for publication.

## Author contributions

Jinli Geng, Data curation, Formal analysis, Validation, Investigation, Writing – review and editing; Yingjun Tang, Data curation, Formal analysis, Methodology, Writing – review and editing; Zhen Yu, Data curation, Investigation, Methodology, Writing – review and editing; Yunming Gao, Data curation, Formal analysis, Validation, Investigation; Wenxiang Li, Data curation, Formal analysis, Investigation, Writing - original draft; Yitong Lu, Bo Wang, Data curation, Formal analysis, Validation; Huiming Zhou, Resources, Methodology; Ping Li, Ping Wang, Resources, Funding acquisition, Writing – review and editing; Nan Liu, Conceptualization, Resources, Writing - original draft, Writing – review and editing; Yubo Fan, Conceptualization, Resources, Funding acquisition, Project administration, Writing – review and editing; Yaxiong Yang, Data curation, Formal analysis, Funding acquisition, Investigation, Writing - original draft, Project administration, Writing – review and editing; Zengcai V Guo, Conceptualization, Resources, Funding acquisition, Investigation, Project administration, Writing – review and editing; Xiaodong Liu, Conceptualization, Resources, Formal analysis, Supervision, Funding acquisition, Investigation, Visualization, Writing - original draft, Project administration, Writing – review and editing

## Author ORCIDs

Yaxiong Yang (ID) http://orcid.org/0000-0002-3313-6049
Zengcai V Guo (ID) http://orcid.org/0000-0002-4140-7961
Xiaodong Liu (ID) http://orcid.org/0000-0002-3171-9611

## Ethics

Procedures involving animals have been approved by local institutional ethical committees (IACUC in Tsinghua University and Beihang University).

## Decision letter and Author response

Decision letter https://doi.org/10.7554/eLife.76691.sa1
Author response https://doi.org/10.7554/eLife.76691.sa2

---

# Additional files

## Supplementary files

• Transparent reporting form

## Data availability

The plasmids of pEGFP-N1-jGCaMP7b-$X_C$ (178361) and pEGFP-N1-jGCaMP7b-$X_N$ (178362) are available on Addgene. Source data for WB and Co-IP are organized as four ZIP files. The data in details associated with the figures have been deposited to Dryad (https://doi.org/10.5061/dryad.zw3r22893).

The following dataset was generated:

| Author(s) | Year | Dataset title | Dataset URL | Database and Identifier |
|---|---|---|---|---|
| Geng J, Tang Y, Yu Z, Yang Y, Guo Z, Liu X | 2022 | Chronic Ca2+ imaging of cortical neurons with long-term expression of GCaMP-X | https://doi.org/10.5061/dryad.zw3r22893 | Dryad Digital Repository, 10.5061/dryad.zw3r22893 |

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
