## [Editor Report]

This paper addresses the toxicity of fluorescent calcium indicators, comparing two series of indicators (GCaMPs and GCaMP-Xs) in mouse cortical neurons. Focusing on calcium oscillations in relation to neuronal morphology, the paper documents GCaMP side effects following prolonged and/or strong expression, and establishes that GCaMP-X indicators are less toxic both in vitro and in vivo. The paper will be of interest to neuroscientists (and others) who use fluorescence calcium indicators for chronic Ca^2+^ imaging.

---

## [Decision Letter]

**Decision letter after peer review:**

Thank you for submitting your article "Chronic ca^2+^ imaging of cortical neurons with long-term expression of GCaMP-X" for consideration by *eLife*. Your article has been reviewed by 2 peer reviewers, and the evaluation has been overseen by a Reviewing Editor and Richard Aldrich as the Senior Editor. The following individual involved in review of your submission has agreed to reveal their identity: Joerg Striessnig (Reviewer #2).

Essential revisions:

1) Provide data showing relative expression levels of the different GCaMP indicators in the study. Such direct measurements of expression levels are needed to rule out the possibility that observed differences in apparent toxicity are not simply explained by variations in protein expression.

2) Reviewer 2 notes the absence of important controls for Figures 2 and 7. For Figure 2, the potential contribution of differences in signal to noise ratios (SNR) of the two indicators should be discussed. For Figure 7, the potential contribution of tTA expression in transgenic neurons inducibly expressing GCaMP6f should be accounted for.

3) Reviewer 1 found the criticisms of GCaMP to be unnecessarily negative and unjustifiably so given that there have been numerous studies using GCaMP under conditions which circumvent the toxicity known to occur when expressed during development. Authors should tone down the anti-GCaMP phrasing and also shorten the Discussion as suggested by reviewer 2.

*Reviewer #1 (Recommendations for the authors):*

While the results are valuable, the authors' conclusions are often unsubstantiated. The authors need to tone down the unnecessary and misleading anti-GCaMP rhetoric and discuss the value of GCaMP-X in the context of neuronal development.

Throughout, the paper suffers from a failure to control critical parameters. The lack of adequate controls risks undermining conclusions about the relative toxicities of GCaMP and GCaMP-X indicators. For example, figure 1 provides compelling evidence that neurons transfected with jGCaMP7 are morphologically distinct from control neurons, and that GCaMP-X-expressing neurons are morphologically similar to control neurons. The difference could be due to the indicator, but there are many other possibilities. Are expression levels of the two indicators comparable, for example? Figure 1 establishes that the cells containing jGCaMP7 were perturbed and those containing GCaMP-X were not, but not that the difference was due to the indicator molecules.

The sensory responsiveness measurements of figure 2 provide another example of missing controls. Responsiveness is measured in GCaMP- and GCaMP-X containing neurons, but not control neurons. Which population is perturbed? And the authors have not accounted for the different SNR characteristics of the two indicators. The evolution of GCaMP indicators provides a lesson in the importance of SNR: with each new GCaMP from GCaMP1 to GCaMP6, a greater proportion of cortical pyramidal neurons appeared sensitive to sensory stimuli, simply because the improved SNR permitted the resolution of smaller signals with each new indicator.

Figure 7 illustrates that neurons from transgenic GCaMP lines are affected in culture. The lines used overexpress tTA and GCaMP, both of which have been implicated in toxicity in the past. Here again, a critical control is missing, leaving the conclusion uncertain.

At what age were virus injections performed? This information is missing from the methods.

Figure 4 appears to be an extension of the morphological analyses in figure 1. Spreading these results over two figures is unnecessary.

There are a large number of supplementary videos that contribute little to the paper. One of two videos would be sufficient.

*Reviewer #2 (Recommendations for the authors):*

The authors should state in the figure legends from how many independent experiments (N) (transections, injected animals, independent cortical cultures) the number of experiments (n) given has been derived from.

Where comparisons between constructs always performed in parallel? That would be state-of the art. Information should be provided in the methods.

"larger voltage-gated activation (VGA)": VGA is a strange term: do they refer to voltage dependence or amplitude? This should be clarified.

Page 19: "SNc neurons switch from HCN channel-based to CaV1.3 channel-based in Parkinson's disease". The role of Cav1.3 channels in SNc neurons for pacemaking is unclear. The Surmeier group has reported in a later paper that Cav1.3 only contributes to pacemaking precision rather than pacemaking frequency (Guzman et al., 2009; PMID 19726659).

---

## [Author Response]

Essential revisions:1) Provide data showing relative expression levels of the different GCaMP indicators in the study. Such direct measurements of expression levels are needed to rule out the possibility that observed differences in apparent toxicity are not simply explained by variations in protein expression.

We performed additional experiments to address this suggestion related to expression levels of GCaMP/GCaMP-X in different settings including cDNA transient transfection, virus infection in live mice, and virus infection in cultured neurons. Here is the summary:

1. We have conducted western-blot and immunocytochemistry experiments to directly measure the expression levels of GCaMP versus GCaMP-X in cultured neurons.

a) The localization tag NES of GCaMP-X_C_ would yield less expression due to space restriction. To compare the potential effects in the cytosol (e.g. on Ca_V_1 channels), the cytosolic (instead of whole-cell/total) expression was focused on.

b) In parallel, for the nuclear effects, the expression level of GCaMP in the nucleus was compared with GCaMP-X_N_.

c) In the original manuscript, the amount of cDNA or virus was under control for fair comparison. Our new data confirmed that the GCaMP expression level in the cytosol was comparable to GCaMP-X_C_.

2. To further rule out the concern, two sets of additional in vitro experiments were conducted:

a) By increasing the amount of GCaMP-X_C_ virus, the total expression was enhanced, to the level comparable to that of GCaMP. Under the above conditions, GCaMP-X_C_ did not exhibit any significant side-effect;

b) In addition, both GCaMP-X_C_ and GCaMP-X_N_ were co-expressed in neurons, leading to higher expression levels than GCaMP, while reaching the same conclusion on the differential performance of GCaMP-X versus GCaMP.

3. Finally, for in vivo imaging, the injected viruses were of differential amount that GCaMP-X was ~10-fold of GCaMP as indicated in the first version. Here in this revision the immunostaining results from brain slices confirmed that the expression level of GCaMP-X was indeed higher than GCaMP.

The relevant data newly acquired include Figure 1—figure supplement 1, Figure 1—figure supplement 2, Figure 4—figure supplement 4 and Figure 6—figure supplement 1. These results are consistent with the design principle that GCaMP-X eliminates unwanted CaM-binding by intramolecular protection regardless of its expression levels. Besides the above new data and evidence, additional information can be found in our earlier report (Yang 2018 PMID: 29666364).

2) Reviewer 2 notes the absence of important controls for Figures 2 and 7. For Figure 2, the potential contribution of differences in signal to noise ratios (SNR) of the two indicators should be discussed. For Figure 7, the potential contribution of tTA expression in transgenic neurons inducibly expressing GCaMP6f should be accounted for.

I. In the context of our work, the effective SNR was calculated mainly to evaluate the overall performance of ca^2+^ imaging. More often, SNR refers to the intrinsic sensing property of the probe as the reviewer mentioned. Examined by artificially generated AP-like waveforms in HEK cells, GCaMP and GCaMP-X resulted in similar SNR values (Yang 2018 PMID: 29666364). Besides the intrinsic SNR, the effective SNR is jointly determined by multiple factors, including the expression level of the probes, and the health state of the neurons. Similar to the original report of GCaMP3 (Tian 2009 PMID: 19898485), the overall SNR could be severely impaired in GCaMPexpressing neurons, presumed to be much better in unaltered/healthy neurons. In this work, the effective/overall SNR using GCaMP-X (of higher dose) turned out to be significantly higher than that from GCaMP, highlighting the advantages of GCaMP-X over GCaMP: healthier neurons while expressing more probes. Meanwhile, we appreciate the review’s insight regarding the retrospect from GCaMP1 to GCaMP6, that the progressive improvements in overall SNR are mostly attributed to enhanced sensing characteristics of the probes. Collectively, a newer GCaMP may improve the overall SNR by enhancing the probe’s sensing characteristics (e.g., GCaMP1-6), or by alleviating neuronal toxicities (e.g., GCaMP-X, or those workaround solutions for GCaMP), or by permitting higher expression levels of probes (e.g. GCaMP-X versus GCaMP). In all, GCaMP-X imaging achieved higher overall SNR than GCaMP, as shown in the revised Figure 2C and Figure 5D.

II. Regarding the contribution of tTA (Tetracyclines) expression in Figure 7, we appreciate the reviewer for pointing it out, which does request additional controls. Indeed, tTA of high level could be toxic to neurons (Moullan 2015 PMID: 25772356) thus a concern for evaluating the toxicity of GCaMP6f. In this revision, the control mouse line of Ai140D was employed as the negative control to compare with Ai148. These two lines are nearly the same except that for Ai140D GFP would be triggered by tTA to express whereas for Ai148 GCaMP6f is to express. Keeping the conditions for these two lines exactly the same, e.g., the same TMP treatment, Ai148 neurons exhibited significantly less neurites than Ai140D neurons. Therefore, our data support that GCaMP but not tTA caused the defects (in comparison with the GFP control group) on neuritogenesis. Additional neurite tracing data demonstrate that no significant difference could be detected due to putative tTA effects, between Ai140D positive neurons and Ai140D negative neurons (or ICR control neurons). In summary, we have reached the same conclusions (as in the original version) regarding GCaM6f effects on neuritogenesis of transgenic Ai148 neurons, after incorporating the new data addressing the potential tTA effects (Figure 7—figure supplement 2).

3) Reviewer 1 found the criticisms of GCaMP to be unnecessarily negative and unjustifiably so given that there have been numerous studies using GCaMP under conditions which circumvent the toxicity known to occur when expressed during development. Authors should tone down the anti-GCaMP phrasing and also shorten the Discussion as suggested by reviewer 2.

We have revised the manuscript according to the suggestions. Meanwhile, we would like to clarify the following matters.

a) We agree that it is fully possible to circumvent the aforementioned problems intrinsic to GCaMP, as extensively demonstrated in numerous reports. Using GCaMP, a great deal of new information/knowledge has been learned and important advances have been achieved. Instead of anti-GCaMP, our GCaMP-X is in line with the efforts from multiple labs to improve GCaMP, in hope to provide our solution regarding its toxicity directly and indirectly associated with unexpected (apo)GCaMP binding in neurons. GCaMP-X is rationally-designed, simple and effective; meanwhile, we agree that the empirical approaches by circumventing the core problem (e.g., limiting the level/time of GCaMP expression) are also effective options. GCaMP probes with the upgrades should continuously contribute to ca^2+^ imaging research, provided that the perturbations of GCaMP could be cautiously controlled.

b) One critic from the reviewer 1 is about the application scope of GCaMP-X to claim based on our data. In this revision, we have clarified that GCaMP-X is an option with multiple virtues for imaging applications involving long-term/high-level GCaMP expressions. Early-stage neurons well represent the scenarios where GCaMP-X and GCaMP could have distinct performances. Together with the new data in this revision, the advantages of GCaMP-X have been confirmed in adult mice or mature neurons. We also agree with the reviewer that neural development should continue to serve as one important theme for future studies to take advantage of GCaMP-X. In addition to the data from adult mice (Figure 2, Figure 5 and Figure 6), the new data/evidence regarding mature/developed neurons can be found in Figure 5—figure supplement 1 and Figure 7—figure supplement 3.

c) Due to its rational-design nature, the improvements of GCaMP-X are in both aspects: in principle (molecular mechanisms) and in practice (cellular and overall effects). In response to the comments/critics, we have revised the text accordingly: to provide the reference of data/figures for each claim when applicable, while avoiding to generalize the “molecular mechanisms” without further/future validations by particular experimental conditions.

d) In fact, there is no perfect probe of zero/null effect on neurons. Even when all other potential effects have been eliminated, GECIs are introduced into the cell acting as exogenous ca^2+^-buffering proteins, which potentially interfere with diverse cellular processes. GCaMP-X has no difference from other probes in the aspect of ca^2+^ buffering effect, thus not a perfect probe either.

e) To provide further support on the design principle, we conducted additional Co-IP experiments to directly examine the interactions between the probes and the apoCaM targets. Under apo conditions, GCaMP-X behaved drastically different from GCaMP, the latter of which bound both α_1D_ (at the IQ domain of the pore forming subunit of Ca_V_1.3 channels) and neurogranin (synaptic protein critical to neuritogenesis specifically binds apoCaM if unphosphorylated). This new set of data can be found in Figure 1—figure supplement 3.

Reviewer #1 (Recommendations for the authors):While the results are valuable, the authors' conclusions are often unsubstantiated. The authors need to tone down the unnecessary and misleading anti-GCaMP rhetoric and discuss the value of GCaMP-X in the context of neuronal development.

We would like to follow the suggestion/critics to carefully and fairly evaluate both probes. We agree with the reviewer that GCaMP-X data in this study have the immediate and direct value to calcium imaging in the context of neuronal development. Meanwhile, GCaMP-X by design is to resolve unwanted molecular interactions of GCaMP (Figure 1 and Figure 1—figure supplement 3), promising a broader scope of applications beyond development. For instance, in our in vivo experiments, high (10fold) dose of GCaMP-X virus was injected to adult mice (82±8 days of age, n=17 if counted by the number of mice) or (~80±6 days, n=28 if counted by injections), yielding higher imaging quality and much less neural toxicity in comparison to GCaMP (Figure 2, Figure 5 and Figure 6). For cultured neurons, new data were acquired from both virus-infected neurons and TMP-induced transgenic neurons, starting to express the probe at the late/mature stages. For these neurons, the comparison between GCaMPX versus GCaMP resulted in similar conclusion (Figure 5—figure supplement 1 and Figure 7—figure supplement 3). In summary, the central problem is the excessive/prolonged expression of GCaMP, which has been resolved by GCaMP-X, thus promising its broad applications, represented by chronic calcium imaging (in vitro and in vivo) in this study.

Besides GCaMP-X, we fully agree that other solutions do exist to circumvent the problem: e.g., to carefully adjust/control the time and level of GCaMP expression as the reviewer pointed out. We have revised the text to explicitly state our overall opinions on GCaMP (see Item 3 in Essential Revisions and also Page 4, Line 107 in the revised manuscript). Meanwhile, we have ensured that the claims regarding GCaMP versus GCaMP-X should not go beyond our testing conditions until further validations.

Throughout, the paper suffers from a failure to control critical parameters. The lack of adequate controls risks undermining conclusions about the relative toxicities of GCaMP and GCaMP-X indicators. For example, figure 1 provides compelling evidence that neurons transfected with jGCaMP7 are morphologically distinct from control neurons, and that GCaMP-X-expressing neurons are morphologically similar to control neurons. The difference could be due to the indicator, but there are many other possibilities. Are expression levels of the two indicators comparable, for example? Figure 1 establishes that the cells containing jGCaMP7 were perturbed and those containing GCaMP-X were not, but not that the difference was due to the indicator molecules.

We agree with the reviewer that more controls would strengthen the conclusions and we appreciate the particular suggestions of the reviewer. We conducted additional experiments for Figure 1 to control the expression levels for transient transfection. Briefly, the difference of GCaMP-X versus GCaMP was not due to the expression level (Figure 1—figure supplement 1 and Figure 1—figure supplement 2). Since the amount of cDNA was controlled, it would be better to separately compare the expression level of GCaMP in the cytosol (versus GCaMP-X_C_) or in the nucleus (versus GCaMP-X_N_). To keep the total expression at the same level, differential amounts of GCaMP versus GCaMP-X_C_ cDNA were transfected into cells. Furthermore, cytosolic GCaMP-X_C_ and nuclear GCaMP-X_N_ were co-expressed in neurons, resulting in the total GCaMP-X proteins at the level comparable to GCaMP. Similar validation experiments were conducted for long-term expression in vitro and in vivo (Figure 4— figure supplement 4 and Figure 6—figure supplement 1). In all these experiments, GCaMP-X consistently outperformed GCaMP, ruling out the concern of expression levels. Please also see our reply to Item 1 in Essential Revisions.

The sensory responsiveness measurements of figure 2 provide another example of missing controls. Responsiveness is measured in GCaMP- and GCaMP-X containing neurons, but not control neurons. Which population is perturbed? And the authors have not accounted for the different SNR characteristics of the two indicators. The evolution of GCaMP indicators provides a lesson in the importance of SNR: with each new GCaMP from GCaMP1 to GCaMP6, a greater proportion of cortical pyramidal neurons appeared sensitive to sensory stimuli, simply because the improved SNR permitted the resolution of smaller signals with each new indicator.

We appreciate the reviewer for pointing out this matter. Additional analyses and experiments on the whisker test have been conducted (see revised Figure 2). First of all, the nucleus-filled neurons (N/C ratio>0.8 as the criteria) exhibited the most severe perturbations. The overall performance of GCaMP-X expressing neurons was significantly better than GCaMP mainly due to the nucleus-filled neurons, manifested by the key indices of amplitude, responsiveness and overall SNR. We agree that the probes’ SNR may also contribute to their difference in overall SNR and responsiveness.

Please see our response (Part I) to Essential Revisions, Item 2.

Figure 7 illustrates that neurons from transgenic GCaMP lines are affected in culture. The lines used overexpress tTA and GCaMP, both of which have been implicated in toxicity in the past. Here again, a critical control is missing, leaving the conclusion uncertain.

We have conducted additional experiments (Figure 7—figure supplement 2) to address the issue of tTA. Briefly, as suggested by the reviewer, an important control group was introduced, i.e., Ai140D mice also based on tTA but expressing GFP instead. The potential complications due to the documented effects of tTA have been excluded.

Please see our response (Part II) to Essential Revisions, Item 2.

At what age were virus injections performed? This information is missing from the methods.

We have systematically collected/verified our records for the timepoint of injection in our in vivo imaging experiments according to the protocol from Dr. Svoboda’s lab (Huber 2012 PMID: 22538608). The injection timepoints in relation to the age of mice have been summarized (82±8 days, n=17 if counted by the number of mice; or ~80±6 days, n=28 if counted by injections), also illustrated in Figure 5—figure supplement 2.

Figure 4 appears to be an extension of the morphological analyses in figure 1. Spreading these results over two figures is unnecessary.

In Figure 1, the morphological analyses were required by the design/validation process of jGCaMP7-X. Following the established procedure (Yang 2018 PMID: 29666364), transient transfection was utilized to quickly (a few days) achieve the proof-of-principle, but not suitable for chronic imaging which was the focus of this study. By virus infection, Figure 4 demonstrated one major aspect of GCaMP-X advantages: less toxicity while enduring prolonged time course (multiple weeks) of probe expression. Figure 4A-C shed new insights toward calcium oscillation dependent neuritogenesis gained in this study (Figure 4D and Figure 4E). In this context, we agree that the morphological analyses for viral delivery of jGCaMP7 and jGCaMP7-X in Figure 1— figure supplement 1 may be more relevant to similar data by GCaMP6 and GCaMP6-X viruses, now changed to Figure 4—figure supplement 3 in this revision.

There are a large number of supplementary videos that contribute little to the paper. One of two videos would be sufficient.

Following this comment and another suggestion from reviewer #2, we have reorganized the supplementary videos as follows:

Figure 3—video supplement 1. Removed.

Figure 3—video supplement 2. Removed.

Figure 3—video supplement 3. Now Figure 3—video 1.

Figure 3—video supplement 4. Removed.

Figure 3—video supplement 5. Removed.

(ultralong lasting Ca^2+^) new Figure 3—video 2.

Figure 3—video supplement 6. Now Figure 3—video 3.

Figure 5—video supplement 1. Removed.

Figure 5—video supplement 2. Now Figure 5—video 1.

Figure 5—video supplement 3. Removed.

Figure 5—video supplement 4. Now Figure 5—video 2.

Figure 7—video supplement 1. Removed.

Figure 7—video supplement 2. Removed.

Figure 8—video supplement 1. Removed.

Reviewer #2 (Recommendations for the authors):The authors should state in the figure legends from how many independent experiments (N) (transections, injected animals, independent cortical cultures) the number of experiments (n) given has been derived from.

We have gone through the figures and legends, by providing the number for each specific set of experiments. Also, in the Methods, our general guidelines for the minimum number of independent experiments have been provided (Page 28, Line 685; Page 30, Line 757; Page 31, Line 769; Page 32, Line 797).

Where comparisons between constructs always performed in parallel? That would be state-of the art. Information should be provided in the methods.

The conditions for GCaMP and GCaMP-X were kept as equal as possible. Briefly, for in vivo two-photon imaging, the left and right brain of the same mouse was injected by GCaMP or GCaMP-X viruses respectively. And for cultured neurons, the groups of GCaMP, GCaMP-X and GFP were maintained under the same conditions, and confocal imaging and other assays were performed within the same DIV to proceed concurrently. Redundant samples were prepared and examined, to exclude any potential artifact if relying on one particular sample, considering the challenges of longterm culturing and imaging. The above information has been included in the revised Methods.

"larger voltage-gated activation (VGA)": VGA is a strange term: do they refer to voltage dependence or amplitude? This should be clarified.

In light of this comment, we decided to keep these details within the caption (instead of the main text). In the original version, VGA referred to voltage-gated activation, adopting the terms previously used in describing Ca_V_1 properties subject to GCaMP perturbations and CaM modulation. In this particular context, both voltage dependence (left-shifted) and current amplitude (increased) were changed, resembling the concurrent effects of CaM over-expression.

Page 19: "SNc neurons switch from HCN channel-based to CaV1.3 channel-based in Parkinson's disease". The role of Cav1.3 channels in SNc neurons for pacemaking is unclear. The Surmeier group has reported in a later paper that Cav1.3 only contributes to pacemaking precision rather than pacemaking frequency (Guzman et al., 2009; PMID 19726659).

We thank the reviewer for this update. We have revised the text on the roles of Ca_V_1.3 in SNc neurons (Page 19 Line 531). Ca^2+^ oscillations are tightly coupled to Ca_V_1.3 in

SNc neurons, as shown in SOP (slow oscillatory potentials) after TTX/Na^+^ blockage.

Dysregulations of Ca_V_1.3 channels or ca^2+^ oscillations are reportedly associated with Parkinson’s disease. Because of the complications regarding the pacemaking mechanisms as pointed out by the reviewer (Guzman 2009 PMID: 19726659), in this revision our discussion focused on Ca_V_1.3-mediated Ca^2+^ signals. The current consensus is that at least Ca_V_1.3 is one of the contributors among others (e.g. HCN) to autonomous SNc activities. Meanwhile, even if not required by pacemaking, Ca_V_1.3 is indispensable for the key tasks at the downstream, e.g., Ca^2+^-dependent dopamine release.